# On Universality Classes of Equivariant Networks

**Marco Pacini**[*]    **Gabriele Santin**[†]    **Bruno Lepri**[‡]    **Shubhendu Trivedi**[§]

## Abstract

Equivariant neural networks provide a principled framework for incorporating symmetry into learning architectures and have been extensively analyzed through the lens of their *separation power*, that is, the ability to distinguish inputs modulo symmetry. This notion plays a central role in settings such as graph learning, where it is often formalized via the Weisfeiler–Leman hierarchy. In contrast, the *universality* of equivariant models—their capacity to approximate target functions—remains comparatively underexplored. In this work, we investigate the approximation power of equivariant neural networks beyond separation constraints. We show that separation power does not fully capture expressivity: models with identical separation power may differ in their approximation ability. To demonstrate this, we characterize the universality classes of shallow invariant networks, providing a general framework for understanding which functions these architectures can approximate. Since equivariant models reduce to invariant ones under projection, this analysis yields sufficient conditions under which shallow equivariant networks fail to be universal. Conversely, we identify settings where shallow models do achieve separation-constrained universality. These positive results, however, depend critically on structural properties of the symmetry group, such as the existence of adequate normal subgroups, which may not hold in important cases like permutation symmetry.

## 1   Introduction

Equivariant neural networks offer a principled framework to incorporate symmetry into learning architectures, attracting sustained attention for both their empirical successes and theoretical richness [1–5]. While their separation power—the capacity to distinguish inputs up to symmetry—has been extensively studied, comparatively less is understood about their approximation capabilities.

In classical approximation theory, expressivity is often characterized via universality—the capacity of a model class to approximate any target function within a given function space to arbitrary precision [6, 7]. In the equivariant setting, however, this notion must be refined. This is because such models treat symmetric inputs as indistinguishable; they can only approximate functions compatible with the underlying symmetry, subject additionally to spurious constraints arising from the imperfect interactions between equivariance and linear inductive biases on neural network layers. In this context, universality becomes inherently relative—defined with respect to a particular separation relation that circumscribes the model's ability to distinguish inputs.

Graph learning has served as a primary testbed for studying invariant and equivariant architectures, where models are typically required to respect node permutation symmetries [4, 8, 9]. Within this setting, separation power is most commonly assessed using the Weisfeiler-Leman (WL) test [10] or homomorphism counting techniques [11]. A whole range of architectures—including Graph

---

[*]University of Trento; Fondazione Bruno Kessler. `mpacini@fbk.eu`

[†]Ca' Foscari University of Venice. `gabriele.santin@unive.it`

[‡]Fondazione Bruno Kessler, `lepri@fbk.eu`

[§]`shubhendu@csail.mit.edu`

39th Conference on Neural Information Processing Systems (NeurIPS 2025).

Neural Networks (GNNs) [12–14], Invariant Graph Networks (IGNs) [4, 15], and subgraph-based models [16, 9]—have been analyzed through this lens. More recently, investigations analyzing separation power have been extended beyond graph-structured data to broader classes of equivariant models [17, 18]. While much of the literature in geometric deep learning has centered on separation as the primary metric of expressivity, recent work [19, 20] has called for a more comprehensive view that includes approximation capabilities more generally. The role of equivariant layers in determining approximation power remains underexplored. Typically, these are composed to increase the model's separation capacity, while a universal component—such as a multilayer perceptron with adjustable width—is appended to approximate functions within the separation-constrained class. As a result, universality is achieved only relative to the distinctions introduced by the equivariant backbone. However, there is no general theory describing how equivariant layers themselves contribute to approximation.

To address this gap, we examine the approximation capabilities of equivariant neural networks beyond what is captured by separation constraints alone. For this purpose, it suffices to focus on invariant architectures, as the core phenomena extend to the equivariant setting. Indeed, projecting the output of an equivariant model onto the trivial representation yields an invariant network. Accordingly, our analysis of invariant networks provides insight into the approximation limits of a broad class of equivariant architectures. We begin by showing that invariant neural networks can be expressed as function that vanishes on certain differential operators (Section 5.1). This formulation allows us to derive sufficient conditions under which a shallow invariant network fails to be universal within the class of separation-constrained continuous functions (Section 5.2).

Our theory and analysis leads to three key insights. First, remarkably, we identify network families that possess identical separation power yet differ in their approximation capabilities—demonstrating that separation alone does not fully characterize expressivity. In particular, we show that shallow networks composed of commonly used equivariant layers—such as PointNets and CNNs with filter width 1—fail to be universal, despite matching the separation power of permutation-invariant continuous functions (Section 6.1). Second, this implies that the only two architectural choices that impact approximation power are depth and the type of hidden representations, the latter being strongly influenced by the structure of the symmetry group. Third, we show that a generalization of the results by [5] produces a broad family of shallow models that are universal within the separation-constrained function class (Section 6.2). However, these constructions fundamentally rely on the structure of the symmetry group. In particular, on the existence of normal subgroups of suitable size, a condition that is not always met, as is the case for key symmetry groups such as the permutation group.

We summarize the main contributions of this work as follows:

- We characterize the universality classes of shallow invariant networks (Theorem 13).

- We establish general *sufficient conditions* under which universality fails, even within function classes exhibiting maximal separation (Theorem 14 and Theorem 15).

- Leveraging these results, we construct explicit examples of invariant models that attain maximal separation yet fail to be universal, demonstrating that separation is not sufficient to guarantee universality (Proposition 16).

- We generalize the results by Ravanbakhsh [5] to a broader family of models (Theorem 18).

## 2   Related Work

Classical approximation theory for neural networks has established foundational results for shallow architectures with sigmoidal activations [6, 7, 21]. Necessary and sufficient conditions on activation functions were later given by Leshno et al. [22], and further refinements appear in Pinkus [23]. For general treatments of approximation theory in modern neural networks, we refer to [24, 25].

Moving beyond shallow networks, Yarotsky [26, 27] proved fundamental results on the approximation rates of deep neural networks, while Siegel [28] derived sharp bounds for deep ReLU networks. These results establish that deep networks are not only universal under mild assumptions but also more parameter-efficient than their shallow counterparts, approximating complex functions with significantly fewer parameters.

Equivariant neural networks offer a principled way to encode symmetry into learning architectures [1–3], with early applications across physics [29], chemistry [30], biology [31], and computer vision [32]. Beyond the foundational work of Yarotsky [33], universality in equivariant and invariant settings has been studied from multiple perspectives. A number of works [5, 34–36] establish universality for certain shallow equivariant networks using unconstrained hidden representations. Keriven and Peyré [37] extended this analysis to equivariant graph neural networks. These proofs rely on two main techniques. The first is the application of the Stone–Weierstrass theorem, or one of its variants, for instance via invariant polynomials [38, 8], to establish density results in spaces of continuous functions. The second is the use of a symmetrization operator, which enforces equivariance but causes the dimension of intermediate representations to grow exponentially. However, these approaches are often impractical: the Stone–Weierstrass theorem cannot be applied directly, since the network families of interest do not form function algebras, while symmetrization leads to prohibitive computational costs. Although canonicalization methods can improve the efficiency of models derived through symmetrization [39, 40], in many cases such techniques cannot be applied [41] or remain computationally inefficient.

A complementary line of work examines permutation-equivariant networks over multisets. Zaheer et al. [42], Qi et al. [43], Segol and Lipman [44] prove universality for such models under constrained hidden representations, but their results are restricted to architectures of depth three. As a result, the universality of truly shallow networks—those with depth two or less—remained unresolved.

In this work, we address this gap and show that certain shallow equivariant networks are *not* universal in the space of equivariant functions. This stands in contrast to the fully connected case, where universality holds generically, and highlights that depth can play a qualitatively different role in equivariant architectures, extending beyond parameter efficiency to approximation capacity itself.

To capture practical models within a theoretical framework, recent work has shifted toward studying universality *up to separation*. In permutation-equivariant networks, expressivity has been analyzed through the Weisfeiler–Leman (WL) hierarchy [45–49], with refinements based on homomorphism counts and subgraph-aware techniques [50, 51]. Joshi et al. [17] extended this approach to geometric domains, deriving depth-sensitive universality results under representation and orbit separation constraints. More generally, Pacini et al. [18] recently characterized the separation power of neural networks for arbitrary finite groups and permutation representations. While these works elucidate distinguishability, they do not fully account for approximation behavior.

The role of equivariant layers in approximation, *beyond* their contribution to separation, remains only partially understood. In practice, such layers are often composed to enhance separation power, followed by a universal component—typically an MLP—to approximate functions within the induced separation class. In the invariant case, this composition can yield universality. In the equivariant case, however, universality is not guaranteed and is only known to hold in specific instances [44]. Thus, the expressive power of the overall model remains fundamentally limited by the separation achieved by the equivariant stack. Yet equivariant layers may also contribute directly to approximation, and in some cases are known to suffice for universality within a fixed separation class [5].

Our work provides a detailed analysis of the universality classes of shallow equivariant networks. We show that equivariant layers are *not always sufficient* to guarantee universality up to separation, and that separation alone is *not a complete proxy* for approximation. We show explicit examples of models with identical separation power but differing approximation capacity. More broadly, we introduce general techniques for comparing the approximation power of equivariant models beyond separation, offering a more refined and complete understanding of expressivity in symmetry-constrained architectures.

# 3 Preliminaries

## 3.1 Groups and Equivariance

We are interested in functions that exhibit symmetry under specified transformations. Mathematically, such symmetries are described by groups: sets of transformations closed under composition, equipped with inverses and an identity element. While group theory offers a rigorous algebraic framework for analyzing symmetry, applying these ideas within neural networks requires their reformulation in linear-algebraic terms. This translation is achieved via representation theory, which associates

abstract group elements with matrix actions on vector spaces. For a brief overview, see Appendix A; for a more detailed treatment, see [52].

Our focus will be on permutation representations, which naturally arise when a group $G$ acts on a finite set $X$. Let $\mathbb{R}^X$ denote the space of real-valued functions on $X$. For each $x \in X$, define $e_x \in \mathbb{R}^X$ as the function taking value 1 at $x$ and 0 elsewhere. The set $\{e_x\}_{x \in X}$ forms a canonical basis for $\mathbb{R}^X$. A permutation representation of $G$ on $V = \mathbb{R}^X$ is a linear action satisfying $g(e_x) = e_{gx}$ for all $g \in G$ and $x \in X$. If $V$ and $W$ are permutation representations of $G$, a map $\phi : V \to W$ is $G$-equivariant if $\phi(gv) = g\phi(v)$ for all $g \in G$ and $v \in V$. We denote by $\mathrm{Hom}(V, W)$ the space of linear maps from $V$ to $W$, and by $\mathrm{Hom}_G(V, W)$ the subspace of $G$-equivariant linear maps. Similarly, let $\mathrm{Aff}(V, W)$ denote the space of affine maps from $V$ to $W$, and $\mathrm{Aff}_G(V, W)$ the subspace of $G$-equivariant affine maps. The spaces $\mathrm{Hom}(V, W)$, $\mathrm{Aff}(V, W)$, and their equivariant counterparts are real vector spaces under pointwise addition and scalar multiplication. A result from Pacini et al. [53] shows that any map $f \in \mathrm{Aff}(V, W)$ admits a unique decomposition of the form $f = \tau_v \circ \phi$ for some $v \in W$ and $\phi \in \mathrm{Hom}(V, W)$, where $\tau_v(w) = w + v$. Such a map is $G$-equivariant iff $\phi$ is $G$-equivariant and $v \in W^G = \{v \in W \mid gv = v; \forall g \in G\}$, the fixed-point subspace of $W$. In particular, there is a linear morphism $\lambda : \mathrm{Aff}_G(V, W) \to \mathrm{Hom}_G(V, W)$ that projects an affine map to its linear part.

## 3.2 Equivariant Neural Networks

With all necessary definitions in place, we now introduce the notion of an equivariant neural network. Throughout this work, we consider networks that are equivariant under the action of a finite group, using arbitrary point-wise continuous activation functions, and with layers that transform according to permutation representations. We adopt the notation introduced by Pacini et al. [18].

**Definition 1** (Point-wise Activation). Let $\sigma : \mathbb{R} \to \mathbb{R}$ be a nonlinear activation function, and let $\mathbb{R}^X$ denote a permutation representation of a group $G$. We define the corresponding *point-wise activation* $\tilde{\sigma} : \mathbb{R}^X \to \mathbb{R}^X$ by setting $\tilde{\sigma}\left(\sum_{x \in X} \alpha_x e_x\right) = \sum_{x \in X} \sigma(\alpha_x) e_x$. When no confusion arises, we will denote both $\sigma$ and $\tilde{\sigma}$ by the same symbol.

**Definition 2** (Neural Networks and Neural Spaces). Let $G$ be a group, and let $V_0, \ldots, V_d$ be permutation representations of $G$. For each $i = 1, \ldots, d$, let $M^i \subseteq \mathrm{Aff}_G(V_{i-1}, V_i)$ be a set of $G$-equivariant affine maps. For $d \geq 2$, the *neural space* associated with the layers $M^1, \ldots, M^d$ and a point-wise activation function $\sigma$ is defined recursively by

$$\mathcal{N}_\sigma(M^1, \ldots, M^d) = \left\{\phi^d \circ \tilde{\sigma} \circ \eta^{d-1} \,\middle|\, \phi^d \in M_d,\ \eta^{d-1} \in \mathcal{N}_\sigma(M^1, \ldots, M^{d-1})\right\},$$

with the base case $\mathcal{N}_\sigma(M^1) = M^1$. An element $\eta^d \in \mathcal{N}_\sigma(M^1, \ldots, M^d)$ is called a *neural network* with layers in $M^1, \ldots, M^d$ and activation $\sigma$. When each $M^i$ is taken to be the full space $\mathrm{Aff}_G(V_{i-1}, V_i)$, we write $\mathcal{N}_\sigma(V_0, \ldots, V_d)$ as shorthand for $\mathcal{N}_\sigma(M^1, \ldots, M^d)$.

To capture architectures commonly used in practice, we adopt a more structured form for the layer spaces $M \subseteq \mathrm{Aff}_G(V, \mathbb{R}^X)$, as proposed in Section 4.2 of Pacini et al. [18]. Specifically, we assume that $M$ takes the form

$$M = \left\{v \mapsto \sum_{i=1}^k x_i \phi^i(v) + \sum_{j=1}^\ell y_j \mathbb{1}_{X_j} \,\middle|\, x_1, \ldots, x_k, y_1, \ldots, y_\ell \in \mathbb{R}\right\}, \tag{1}$$

where $\phi^1, \ldots, \phi^k$ span a subspace of $\mathrm{Hom}_G(V, \mathbb{R}^X)$, $X_1, \ldots, X_\ell$ are the orbits of $X$ under the $G$-action, and $\mathbb{1}_{X_i} := \sum_{x \in X_i} e_x$ for $i = 1, \ldots, \ell$. This formulation, while notation-heavy, plays a central role in the development of our main results.

We now present two working examples of equivariant affine maps and their associated neural spaces. These examples both reflect architectures commonly used in geometric deep learning and illustrate how standard models naturally conform to the structure in (1). They will serve as recurring reference points throughout to highlight key phenomena in the universality landscape of equivariant networks.

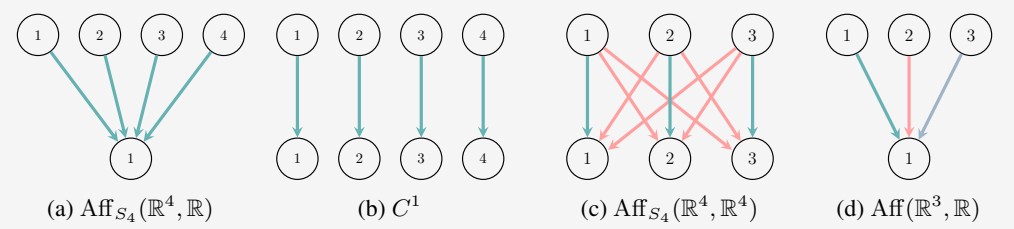

(a) $\mathrm{Aff}_{S_4}(\mathbb{R}^4, \mathbb{R})$   (b) $C^1$   (c) $\mathrm{Aff}_{S_4}(\mathbb{R}^4, \mathbb{R}^4)$   (d) $\mathrm{Aff}(\mathbb{R}^3, \mathbb{R})$

Figure 1: Colored bipartite graphs illustrating the layer spaces used in Examples 3 and 4, along with the graph corresponding to a standard fully connected layer, following the convention of Ravanbakhsh et al. [54] where arrows of the same color denote identical weights applied to different input values.

**Example 3** (PointNets). *We focus on the sum-pooling variant of PointNet architectures [43], which are designed to process unordered collections, such as point clouds, by enforcing permutation equivariance. An input configuration of $n$ elements with $f$-dimensional features is represented by a tensor $A \in \mathbb{R}^{n \times f}$, where each row corresponds to the features of a single object. Permuting the elements corresponds to permuting the rows of $A$, i.e., the indices along its first axis. In our framework, the input tensor $A$ is modeled as an element of $\mathbb{R}^X \otimes \mathbb{R}^f$, where $X = [n]$ and the symmetric group $G = S_n$ acts on $X$ via its standard action and trivially on $\mathbb{R}^f$. PointNet architectures operate on such inputs using layers in the space $\mathrm{Aff}_{S_n}(\mathbb{R}^X \otimes \mathbb{R}^{f_{i-1}}, \mathbb{R}^X \otimes \mathbb{R}^{f_i})$, where each $\mathbb{R}^{f_i}$ corresponds to a space of $S_n$-invariant hidden features. Accordingly, the neural spaces corresponding to these equivariant architectures and their invariant counterparts take the following forms, respectively:*

$$\mathcal{N}_\sigma(\mathbb{R}^{X_0} \otimes \mathbb{R}^{f_0}, \dots, \mathbb{R}^{X_d} \otimes \mathbb{R}^{f_d}) \quad \text{and} \quad \mathcal{N}_\sigma(\mathbb{R}^{X_0} \otimes \mathbb{R}^{f_0}, \dots, \mathbb{R}^{X_{d-1}} \otimes \mathbb{R}^{f_{d-1}}, \mathbb{R}^{f_d}).$$

*Zaheer et al. [42] showed that understanding the structure of $\mathrm{Aff}_{S_n}(\mathbb{R}^X \otimes \mathbb{R}^{f_{i-1}}, \mathbb{R}^X \otimes \mathbb{R}^{f_i})$ reduces to understanding $\mathrm{Aff}_{S_n}(\mathbb{R}^X, \mathbb{R}^X)$. Identifying $\mathbb{R}^X$ with $\mathbb{R}^n$, they established that*

$$\mathrm{Aff}_{S_n}(\mathbb{R}^n, \mathbb{R}^n) = \left\{ v \mapsto (x_1 \, \mathrm{id} + x_2 \, \mathbb{1}\mathbb{1}^\top)v + y\mathbb{1} \mid x_1, x_2, y \in \mathbb{R} \right\},$$

*where $\mathbb{1} = \mathbb{1}_{[n]} = [1, \dots, 1]^\top$. Figure 1b shows the colored bipartite graph corresponding to the layer space $\mathrm{Aff}_{S_n}(\mathbb{R}^n, \mathbb{R}^n)$. In the invariant case, $\mathrm{Aff}_{S_n}(\mathbb{R}^n, \mathbb{R}) = \left\{ v \mapsto x \, \mathbb{1}^\top v + y \mid x, y \in \mathbb{R} \right\}$, which is consistent with the notation introduced in (1). Figure 1a shows the colored bipartite graph corresponding to the layer space $\mathrm{Aff}_{S_n}(\mathbb{R}^n, \mathbb{R})$.*

**Example 4** (Convolutional Neural Networks). *Circular convolutional filters can be naturally formulated within the framework of permutation representations. For simplicity, we focus on the one-dimensional case. Let $X = [n]$ and let $G = \mathbb{Z}_n$ act on $X$ by modular shifts. Identifying $\mathbb{R}^X$ with $\mathbb{R}^n$, the space $\mathrm{Hom}_{\mathbb{Z}_n}(\mathbb{R}^n, \mathbb{R}^n)$ corresponds to circulant matrices $A(x)$, each determined by a generating vector $x = (x_1, \dots, x_n) \in \mathbb{R}^n$, as shown below.*

*Each map in $\mathrm{Aff}_{\mathbb{Z}_n}(\mathbb{R}^n, \mathbb{R}^n)$ consists of a linear part defined by a circulant matrix and a bias term in $\mathbb{R}^n$:*

$$A(x) := \begin{bmatrix} x_1 & x_n & x_{n-1} & \cdots & x_2 \\ x_2 & x_1 & x_n & \cdots & x_3 \\ x_3 & x_2 & x_1 & \cdots & x_4 \\ \vdots & \vdots & \vdots & \ddots & \vdots \\ x_n & x_{n-1} & x_{n-2} & \cdots & x_1 \end{bmatrix} \text{ and } y\mathbb{1}_X = y\mathbb{1}_{[n]} = y \begin{bmatrix} 1 \\ \vdots \\ 1 \end{bmatrix}.$$

*Observe that any circulant matrix $A(x)$ can be written as a linear combination $A(e_1), \dots, A(e_n)$, that is, $A(x) = \sum_{i=1}^n x_i A(e_i)$ where $\{e_1, \dots, e_n\}$ denotes the standard basis of $\mathbb{R}^n$. Since limited-width convolutional filters are standard in practice, we restrict attention to the following maps:*

$$C^k = \left\{ v \mapsto \sum_{i=1}^k x_i A(e_i)v + y\mathbb{1}_{[n]} \mid x_1, \dots, x_k, y \in \mathbb{R} \right\}. \tag{2}$$

*This class can be seen as the one-dimensional analogue of the $k \times k$ convolutional kernels widely used in 2-D computer vision applications. The corresponding neural space is given by $\mathcal{N}_\sigma(C^{k_1}, \dots, C^{k_d})$, for a choice of filter sizes $1 \le k_1, \dots, k_d \le n$. Circular invariant layers can be characterized as*

$$I := \mathrm{Aff}_{\mathbb{Z}_n}(\mathbb{R}^n, \mathbb{R}) = \left\{ v \mapsto (x \, \mathbb{1}^\top) \cdot v + y \mid x, y \in \mathbb{R} \right\}. \tag{3}$$

In particular, we focus on the spaces $C^1$, which correspond to convolutional filters of width one; see Figure 1c for the colored bipartite graph representing the space $\text{Aff}_{S_n}(\mathbb{R}^n, \mathbb{R}^n)$.

Having detailed the structure of the layer spaces and their correspondence to practical architectures, we now turn to the study of universality in families of shallow neural spaces.

## 4 Universality in Shallow Neural Spaces

**Universality Classes.** To establish notation and introduce the notion of universality classes, we begin by reformulating the classical universality result for shallow neural networks [23] in terms of our framework. Observe that the full class of shallow neural networks with variable width can be written as $\bigcup_{h \in \mathbb{N}} \mathcal{N}_\sigma(\mathbb{R}^m, \mathbb{R}^h, \mathbb{R})$. We denote by $\mathcal{U}_\sigma(\mathbb{R}^m, \mathbb{R}, \mathbb{R})$ the associated universality class—namely, the set of continuous functions on $\mathbb{R}^m$ approximable by such networks. Formally, $\mathcal{U}_\sigma(\mathbb{R}^m, \mathbb{R}, \mathbb{R})$ is defined as the closure of this union in $\mathcal{C}(\mathbb{R}^m)$, equipped with the topology of uniform convergence on compact sets.

**Theorem 5.** *The universality class for shallow neural networks, $\mathcal{U}_\sigma(\mathbb{R}^m, \mathbb{R}, \mathbb{R})$, coincides with $\mathcal{C}(\mathbb{R}^m)$ if and only if the activation function $\sigma$ is not a polynomial.*

An analogous result in the equivariant setting was established by Ravanbakhsh [5] for neural networks defined on representations $V$ and $W$ as input and output spaces, respectively, and with regular hidden representations of the form $\mathbb{R}^G$. We define the universality class $\mathcal{U}_\sigma(V, \mathbb{R}^G, W)$ as the set of functions in $\mathcal{C}(V, W)$ that can be approximated by elements of $\bigcup_{h \in \mathbb{N}} \mathcal{N}_\sigma(V, \mathbb{R}^G \otimes \mathbb{R}^h, W)$. Note that, in analogy with classical networks, the role of width is played by the hyperparameter $h$, which determines the dimension of the invariant hidden representation. The results of Ravanbakhsh [5] can then be stated as follows.

**Theorem 6.** *The universality class $\mathcal{U}_\sigma(V, \mathbb{R}^G, W)$ coincides with $\mathcal{C}_G(V, W)$, the space of continuous $G$-equivariant functions from $V$ to $W$, if and only if the activation function $\sigma$ is not a polynomial.*

We aim to provide a definition of universality classes that encompasses the notions introduced in Theorem 5 and Theorem 6, while being general enough to cover a wider range of architectures, such as PointNets and CNNs with variable filter size. To this end, we introduce the following auxiliary notation. Let $G$ be a finite group and let $V$, $W$ and $Z$ be permutation representations of $G$. Let $M$ be a subspace of $\text{Aff}_G(V, W)$ and $N$ a subspace of $\text{Aff}_G(W, Z)$, as defined in (1). Then, for each $h \in \mathbb{N}$, we define $M_h$ as the subspace of $\text{Aff}_G(V, W \otimes \mathbb{R}^h)$ given by

$$M_h := \{x \mapsto (f_1(x), \ldots, f_h(x)) \mid f_1, \ldots, f_h \in M\}, \tag{4}$$

and define $_hN$ as the subspace of $\text{Aff}_G(W \otimes \mathbb{R}^h, Z)$ given by

$$_hN := \{(x_1, \ldots, x_h) \mapsto g_1(x_1) + \cdots + g_h(x_h) \mid g_1, \ldots, g_h \in N\}. \tag{5}$$

Recalling the isomorphism $W \otimes \mathbb{R}^h \cong (W)^{\oplus h}$, note that in the special cases where $M = \text{Aff}_G(V, W)$ and $N = \text{Aff}_G(W, Z)$, we have $M_h \cong \text{Aff}_G(V, W \otimes \mathbb{R}^h)$ and $_hN \cong \text{Aff}_G(W \otimes \mathbb{R}^h, Z)$. With this notation in place, we can now provide a general definition of universality classes.

**Definition 7** (Universality Classes)**.** The universality class $\mathcal{U}_\sigma(M, N)$ associated with a family of neural spaces $\mathcal{N}_\sigma(M_{h,h} N)$ for $h \in \mathbb{N}$ is the set of continuous functions approximated by these neural networks. More formally, $\mathcal{U}_\sigma(M, N)$ is defined as the closure of $\bigcup_{h \in \mathbb{N}} \mathcal{N}_\sigma(M_{h,h} N)$ in $\mathcal{C}(V, Z)$, equipped with the topology of uniform convergence on compact sets. As in the case of neural spaces, when $M = \text{Aff}_G(V, W)$ and $N = \text{Aff}_G(W, Z)$, we will simply write $\mathcal{U}_\sigma(V, W, Z)$.

However, comparing different universality classes is particularly challenging, and in the literature, separation power has often been used as a proxy for this purpose. The next section revisits this notion and critically examines its adequacy as a surrogate for universality.

**On Separation-Constrained Universality**. Theorem 6 establishes that equivariant neural networks cannot approximate all continuous functions. In particular, invariant networks are inherently unable to distinguish between symmetric inputs—a limitation that naturally constrains the class of functions they can represent. To make this precise, we formally define the notion of separation and the concept of *separation-constrained universality*.

**Definition 8** (Separation-Constrained Universality). A family of functions $\mathcal{N} \subseteq \{f : X \to Y\}$ is said to *separate* $\alpha$ and $\beta$ if there exists $f \in \mathcal{N}$ such that $f(\alpha) \neq f(\beta)$. The set of point pairs not separated by $\mathcal{N}$ defines an equivalence relation:

$$\rho(\mathcal{N}) = \{(\alpha, \beta) \in X \times X \mid f(\alpha) = f(\beta) \text{ for all } f \in \mathcal{N}\}.$$

A family $\mathcal{N}$ is said to be *separation-constrained universal* if its relative universality class coincides with the entire set of continuous functions that respect the equivalence relation $\rho(\mathcal{N})$, that is,

$$\mathcal{C}_\rho(X, Y) = \{f \in \mathcal{C}(X, Y) \mid f(x) = f(y) \text{ for all } (x, y) \in \rho(\mathcal{N})\}.$$

It is a standard fact in approximation theory [55] that if a family of functions $\mathcal{N}$ fails to separate two points, then it cannot approximate any function that does. As such, separation-constrained universality captures the maximal expressivity achievable by $\mathcal{N}$. Here, we aim to investigate whether separation alone suffices to characterize expressivity i.e., whether universality classes with the same separation power must necessarily coincide. To this end, we now present three network families that share the same separation relation, despite differing in their internal representations. Throughout the remainder of the paper, we assume that all activation functions $\sigma : \mathbb{R} \to \mathbb{R}$ are non-polynomial.

**Proposition 9.** [5] *Let $C^1$ be defined as in (2), representing convolutional filters of width $1$, and let $I$ be as defined as in (3), representing invariant circular layers. Let $S_n$ act on $\mathbb{R}^n \cong \mathbb{R}^{[n]}$ via the standard permutation action. Then, the following universality classes have the same separation power:*

$$\rho\left(\mathcal{U}_\sigma(C^1, I)\right) = \rho\left(\mathcal{U}_\sigma(\mathbb{R}^n, \mathbb{R}^n, \mathbb{R})\right) = \rho\left(\mathcal{U}_\sigma(\mathbb{R}^n, \mathbb{R}^{S_n}, \mathbb{R})\right).$$

This naturally raises the following question.

**Question 10.** *Are these universality classes equal as well? More generally, is separation a complete proxy for comparing universality classes?*

We answer Question 10 in the negative via Proposition 16, after developing the necessary theory.

## 5 Main Results

In this section, we characterize the universality classes of invariant shallow neural networks (Section 5.1) and compare them (Section 5.2). Although the characterization is restricted to the invariant case, the following remark shows that it can be used to demonstrate that non-approximation in the invariant setting implies failure in the equivariant case as well.

*Remark* 11. Let $\mathcal{U}_1 \subseteq \mathcal{U}_2 \subseteq \mathcal{C}_G(V, W)$ be two universality classes with input space $V$ and output space $W$. Let $\pi : W \to W^G$ denote the projection onto the trivial component of $W$. Define the pullback map

$$\pi^* : \begin{array}{ccc} \mathcal{C}_G(V, W) & \longrightarrow & \mathcal{C}_G(V, W^G) \\ f & \longmapsto & \pi \circ f, \end{array}$$

where $\mathcal{C}_G(V, W)$ denotes the space of continuous equivariant functions from $V$ to $W$. Then, $\pi^*(\mathcal{U}_1) \subsetneq \pi^*(\mathcal{U}_2)$ implies $\mathcal{U}_1 \subsetneq \mathcal{U}_2$, since $\pi^*$ is a continuous linear operator. This shows that a sufficient condition for strict inclusion between spaces of invariant networks also yields a sufficient condition for strict inclusion between the corresponding spaces of equivariant networks.

With this observation, we now restrict our attention to invariant networks without loss of generality.

### 5.1 Characterization of Universality Classes

To characterize the universality classes of invariant shallow networks, we begin by introducing the notion of a *basis map*.

**Definition 12** (Basis maps). As defined in (1), let $M$ be a subspace of $\mathrm{Aff}_G(V, \mathbb{R}^Y)$, where $V$ is a permutation representation and $Y$ is a finite $G$-set of cardinality $\ell$, which we identify with $[\ell]$. Let $\phi^1, \ldots, \phi^m$ be a basis for the linear part of $M$, and for each $i \in Y$, define the linear maps

$$\phi_i : \begin{array}{c} \mathbb{R}^X \to \mathbb{R}^m \\ x \mapsto (\phi_i^1(x), \ldots, \phi_i^m(x)). \end{array} \tag{6}$$

We refer to the maps $\phi_1, \ldots, \phi_\ell$ as the *basis maps* associated with $M$ or its basis $\phi^1, \ldots, \phi^m$.

---

[5]For clarity of presentation, all proofs are deferred to the Appendix, with the exception of Proposition 16.

We now state the central characterization theorem for universality classes in terms of differential constraints on invariant functions.

**Theorem 13.** *Let $M$ and $N$ be, respectively, subspaces of $\mathrm{Aff}_G(V, W)$ and $\mathrm{Aff}_G(W, \mathbb{R})$. Let $f$ be an invariant function, then $f \in \mathcal{U}_\sigma(M, N)$ if and only if $P(\partial_1, \ldots, \partial_d)f = 0$ for every polynomial $P$ that vanishes on the spaces spanned by the rows $\phi_i^1, \ldots, \phi_i^m$ of each basis map $\phi_1, \ldots, \phi_\ell$, see* (6).

Here, we assume $d = \dim V$, and let $P(\partial_1, \ldots, \partial_d)$ denote the constant-coefficient linear differential operator associated with the polynomial $P$. The derivatives $\partial_i$ on $V$ are interpreted in the distributional sense; see [56] for details.

Although Theorem 13 provides a complete characterization of the universality classes for arbitrary families of neural spaces, this generality may come at the cost of practicality. Indeed, computing the exact set of polynomials $P$ can be particularly challenging, due to the combinatorial complexity arising from the intersections of the subspaces spanned by $\phi_i^1, \ldots, \phi_i^m$. Nonetheless, the theorem is not merely of theoretical interest—it plays a central role in deriving sufficient conditions for universality failure. These conditions enable a principled comparison of the approximation power of distinct model families, as we explore in the following sections.

## 5.2 Sufficient Conditions for Universality Failure

In this section, we present two sufficient conditions for the failure of separation-constrained universality. These results will be used to resolve Question 10 and to prove Proposition 16. We begin with Theorem 14, which provides a general—but more difficult to verify—criterion, followed by Theorem 15, a less general version that is simpler to apply, despite its more convoluted appearance.

First, we introduce the notion of a directional derivative. For each vector $c = (c_1, \ldots, c_n) \in \mathbb{R}^n$, the directional derivative is defined as the differential operator $D_c = c_1 \cdot \partial_1 + \cdots + c_n \cdot \partial_n$.

**Theorem 14.** *A continuous function $f$ does not belong to the class $\mathcal{U}_\sigma(M, N)$ if*

$$D_{c_1} \cdots D_{c_\ell} f \neq 0 \tag{7}$$

*for some choice of $c_\alpha$ in $\ker(\phi_\alpha^\top)$ for each basis map $\phi_1, \ldots, \phi_\ell$.*

In the case of equivariant networks where each affine layer is allowed to be an arbitrary equivariant affine map, Theorem 14 can be strengthened as follows.

**Theorem 15.** *Let $M = \mathrm{Aff}_G(V, W)$ and $N = \mathrm{Aff}_G(W, \mathbb{R})$, where $V$ and $W$ are permutation representations. Let $\phi_1, \ldots, \phi_\ell$ denote the basis maps associated with $M$, see* (6). *Then, the universal class $\mathcal{U}_\sigma(M, N)$ fails to be separation-constrained universal if, for some choice of:*

- *integers $s_1, \ldots, s_\ell \in \{0, \ldots, \ell\}$ satisfying $s_1 + \cdots + s_\ell = \ell$,*

- *integers $a_1 > \ell$ and $a_i + \ell < a_{i+1}$ for each $i = 1, \ldots, \ell$,*

- *vectors $c_i \in \ker(\phi_i^\top)$ for each $i = 1, \ldots, \ell$,*

*Let $i_1, \ldots, i_r$ be the indices such that $s_{i_j} \neq 0$. The following expression is nonzero:*

$$\sum_{\sigma \in S_\ell} \frac{a_{i_1}!}{s_{i_1}!} \cdots \frac{a_{i_r}!}{s_{i_r}!}(c_{\sigma(1),1} \cdots c_{\sigma(s_1),1}) \cdot (c_{\sigma(s_1+1),2} \cdots c_{\sigma(s_1+s_2),2}) \cdots (c_{\sigma(\ell-s_\ell),\ell} \cdots c_{\ell,\ell}).$$

# 6 The Heterogeneous Landscape of Universality Classes

We now apply the tools developed in Section 5 to investigate the structure of universality classes and illustrate their heterogeneity. In Section 6.1, we address Question 10 by applying Theorems 14 and 15 to exhibit concrete examples of failure. In contrast, Section 6.2 presents Theorem 18, a generalization of Theorem 6, which provides sufficient conditions for achieving separation-constrained universality—highlighting the diversity of behaviors even within fixed symmetry classes.

## 6.1 Examples of Failure

**Proposition 16.** *As established in Proposition 9, the following spaces achieve the same separation power, yet differ in their approximation capabilities when $n > 2$:*

$$\mathcal{U}_\sigma(C^1, I) \subsetneq \mathcal{U}_\sigma(\mathbb{R}^n, \mathbb{R}^n, \mathbb{R}) \subsetneq \mathcal{U}_\sigma(\mathbb{R}^n, \mathbb{R}^{S_n}, \mathbb{R}).$$

*By Remark 11, the corresponding equivariant models also have distinct approximation power.*

We will prove the two strict inclusions of Proposition 16 in the following three paragraphs.

**Failure for CNN with filter width 1:** We now apply Theorem 14 to show that CNNs with filter width 1 cannot approximate the function $(x_1 + \cdots + x_n)^n$ for $n > 1$, namely $(x_1 + \cdots + x_n)^n \notin \mathcal{U}_\sigma(C^1, I)$. Indeed, for any $\alpha = 1, \ldots, n$, we have $e_{\alpha+1} \in \ker(\pi_\alpha^\top) = \mathrm{Span}\{e_1, \ldots, \hat{e}_\alpha, \ldots, e_n\}$, where $\alpha + 1$ is modulo $n$. Moreover, note that $D_{e_\alpha} = \partial_\alpha$, thus $\partial_n \cdots \partial_1 (x_1 + \cdots + x_n)^n = n! \neq 0$, which violates (7) in Theorem 14.

**Success for PointNet:** We now show that shallow PointNets approximate the polynomial function $(x_1 + \cdots + x_n)^n$. By Proposition 41 in Appendix D, $f(x_1, x_1 + \cdots + x_n) + \cdots + f(x_n, x_1 + \cdots + x_n)$ belongs to $\mathcal{U}_\sigma(\mathbb{R}^n, \mathbb{R}^n, \mathbb{R})$ for any $f \in \mathcal{C}(\mathbb{R}^2)$. In particular, for $f(x, y) := y^n \in \mathcal{C}(\mathbb{R}^2)$, we see that $(x_1 + \cdots + x_n)^n \in \mathcal{U}_\sigma(\mathbb{R}^n, \mathbb{R}^n, \mathbb{R})$. Together with the previous observation, this establishes the first strict inclusion in Proposition 16, namely $\mathcal{U}_\sigma(C^1, I) \subsetneq \mathcal{U}_\sigma(\mathbb{R}^n, \mathbb{R}^n, \mathbb{R})$.

**Failure for PointNet:** We now aim to show that shallow PointNets cannot approximate the polynomial function $x_1 \cdots x_n$, which is $S_n$-invariant and therefore should, in principle, be approximable in a separation-constrained setting. We distinguish two cases: $n > 3$ and $n = 3$. Note that for $n = 2$, the symmetric group $S_2$ is abelian, and universality follows directly from Theorem 6.
We start considering $(n > 3)$. We again employ Theorem 14 to show that shallow invariant PointNets cannot approximate $x_1 \cdots x_n$, and hence neither CNNs with filter width 1. Indeed, note that the basis maps for $\mathrm{Aff}_{S_n}(\mathbb{R}^n, \mathbb{R}^n)$ in this case are given by $\phi_\alpha(x_1, \ldots, x_n) = (x_\alpha, x_1 + \cdots + x_n)$. In matrix form, we write $\phi_\alpha = [e_\alpha, \mathbb{1}]^\top$. We define $K_\alpha := \ker(\phi_\alpha^\top) = \mathrm{Span}(e_i - e_j)_{i,j=1,\ldots,\hat{\alpha},\ldots,n}$. Then, define the following direction vectors:

$$c_1 := e_2 - e_n \in K_1, \qquad c_2 := e_3 - e_n \in K_2,$$

$$\vdots$$

$$c_{n-3} := e_{n-2} - e_n \in K_{n-3}, \qquad c_{n-2} := e_{n-1} - e_n \in K_{n-2},$$
$$c_{n-1} := e_n - e_2 \in K_{n-1}, \qquad c_n := e_1 - e_2 \in K_n.$$

Explicit computation shows that $D_{c_n} \cdots D_{c_1}(x_1 \cdots x_n) = 2$, verifying (7).
The previous technique does not apply in the case $n = 3$, for which we must instead resort to Theorem 15. First, define $c_1 := e_2 - e_3 \in K_1$, $c_2 := e_3 - e_1 \in K_2$, and $c_3 := e_1 - e_2 \in K_3$. Note that $c_{i,i} = 0$ for each $i = 1, 2, 3$. For $s_1 = 2$, $s_2 = 1$, and $s_3 = 0$, the polynomial becomes $a_1(a_1 - 1)a_2 \cdot [c_{3,1} \cdot c_{2,1} \cdot c_{1,2}] = -a_1(a_1 - 1)a_2 \neq 0$ by choosing $a_1, a_2 > 3$.

In view of the universality results for PointNet with depth 3 and arbitrary widths in both hidden layers by Segol and Lipman [44], this example highlights how, in the case of permutation equivariance, depth is crucial for achieving separation-constrained universality. This contrasts with other settings where universality can be achieved without relying on depth, as we will describe in the next section.

## 6.2 Examples of Separation-Constrained Universality

We now present Theorem 18, a generalization of Theorem 6, which shows that a specific class of hidden representations can achieve separation-constrained universality. These representations arise from cosets of particular subgroups $H$ of $G$, defined as follows:

**Definition 17** (Normal subgroup). A subgroup $H$ is *normal* if $ghg^{-1} \in H$ for each $h \in H, g \in G$.

**Theorem 18.** *Let $V$ and $Z$ be permutation representations of a finite group $G$, and let $H$ be a normal subgroup of $G$. Therefore, $\mathcal{U}_\sigma(V, \mathbb{R}^{G/H}, Z)$ is separation-constrained universal.*

The converse does not always hold: representations arising from non-normal subgroups may nevertheless achieve separation-constrained universality, as illustrated in the following remark.

*Remark* 19. Let $H$ be a non-normal subgroup of $S_n$ contained in $A_n$. Then

$$\mathcal{U}_\sigma(\mathbb{R}^{S_n/A_n}, \mathbb{R}^{S_n/A_n}, \mathbb{R}) = \mathcal{U}_\sigma(\mathbb{R}^{S_n/A_n}, \mathbb{R}^{S_n/H}, \mathbb{R}) = \mathcal{C}_{S_n}(\mathbb{R}^{S_n/A_n}).$$

All subgroups of an abelian group are normal, whereas $S_n$ has only one non-trivial normal subgroup, $A_n$, with $|S_n/A_n| = 2$, yielding hidden representations that are too small to be effective. We summarize by noting that intermediate representations built from abelian groups, such as those in standard circular CNNs, achieve separation-constrained universality. In contrast, architectures based on permutation representations lack this guarantee, as shown in Proposition 16.

## 7 Limitations

This work represents a first step toward understanding the approximation capabilities of equivariant networks beyond separation. Several limitations, however, remain. In particular, our analysis is limited to shallow networks. While these serve as minimal and analytically tractable examples, they may not fully capture the behavior of deeper architectures. Extending this framework to deeper networks—particularly in settings where depth interacts nontrivially with separation, as in IGNs—poses a significant challenge.

## 8 Conclusions

We investigated the approximation capabilities of equivariant neural networks, moving beyond their well-studied separation properties. By formulating shallow invariant networks as generalized superpositions of ridge functions (see Proposition 41), we developed a novel characterization of their universality classes and examined how architectural choices influence approximation behavior. Our analysis reveals that even networks with maximal separation power may fail to approximate all functions within the corresponding symmetry-respecting class, a phenomenon we attribute to the structure of their hidden representations. These findings suggest that approximation power cannot be deduced from separation alone and should be treated as a distinct axis of expressivity. Our results thus call for a more nuanced understanding of equivariant architectures—one that takes both axes into account in theoretical analysis and model design.

As future directions, we aim to extend our framework to determine whether failures of separation-constrained universality, such as those established in Proposition 16, persist in deeper architectures. Another important avenue for investigation is how differences in expressivity affect generalization, particularly among models that share the same separation power.

## 9 Acknoledgements

Bruno Lepri acknowledges the support of the PNRR project FAIR - Future AI Research (PE00000013), under the NRRP MUR program funded by the NextGenerationEU and the support of the European Union's Horizon Europe research and innovation program under grant agreement No. 101120237 (ELIAS). This work was also partly supported by Ministero delle Imprese e del Made in Italy (IPCEI Cloud DM 27 giugno 2022 – IPCEI-CL-0000007) and European Union (Next Generation EU).

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

# A  Preliminaries

## A.1  Group Theory

**Definition 20** (Group). A *group* is a set $G$ equipped with a binary operation $\cdot : G \times G \to G$ satisfying the following properties:

- **Associativity:** for all $g, h, k \in G$, we have $(g \cdot h) \cdot k = g \cdot (h \cdot k)$.

- **Identity element:** there exists an element $e \in G$ such that $g \cdot e = e \cdot g = g$ for every $g \in G$.

- **Inverses:** for every $g \in G$, there exists an element $g^{-1} \in G$ such that $g \cdot g^{-1} = g^{-1} \cdot g = e$.

The group is said to be *finite* if $G$ contains finitely many elements. It is called *abelian* (or *commutative*) if $g \cdot h = h \cdot g$ holds for all $g, h \in G$.

We now define the concept of a group homomorphism, a structure-preserving map between groups.

**Definition 21** (Homomorphism). Let $G$ and $H$ be groups. A function $\phi : G \to H$ is called a *group homomorphism* if, for all $g, h \in G$, it holds that
$$\phi(g \cdot h) = \phi(g) \cdot \phi(h).$$

**Definition 22** (Cosets). Let $G$ be a group and let $H \leq G$ be a subgroup. The *left coset* of $H$ associated with an element $g \in G$ is the set
$$gH = \{gh \mid h \in H\}.$$
The collection of all left cosets of $H$ in $G$ is denoted by $G/H = \{gH \mid g \in G\}$.

Similarly, the *right coset* of $H$ corresponding to $g \in G$ is defined as
$$Hg = \{hg \mid h \in H\},$$
and the set of all left cosets is written as $G/H = \{gH \mid g \in G\}$.

Given another subgroup $K \leq G$, the *double coset* associated with $g \in G$ is the set
$$HgK = \{hgk \mid h \in H, \, k \in K\},$$
and the set of all such double cosets is denoted by $H\backslash G/K$.

**Definition 23** (Normal subgroup). A subgroup $H$ is *normal* if $ghg^{-1} \in H$ for each $h \in H, g \in G$.

**Example 24.** *We highlight two families of normal subgroups relevant to our discussion:*

1. *All subgroups of abelian groups are normal.*

2. *The alternating group $A_n$ is the only non-trivial normal subgroup of $S_n$.*

**Theorem 25.** *If $H$ is a normal subgroup of $G$, then the cosets $G/H$ for a group, where the binary operation is defined as $g_1 H \cdot g_2 H = g_1 g_2 H$.*

## A.2  Group Actions and Equivariance

Let $G$ be a group and let $X$ be a set. A *group action* of $G$ on $X$ is a map
$$\Phi : G \times X \to X,$$
commonly written as $\phi_g(x) = \Phi(g, x)$ for $g \in G$ and $x \in X$, that satisfies the following two conditions:

- **Identity:** $\phi_e = \mathrm{id}_X$, where $e$ is the identity element in $G$.
- **Compatibility:** For all $g, h \in G$, we have $\phi_g \circ \phi_h = \phi_{gh}$.

In practice, we often denote the action by $g \cdot x$ or simply $gx$ in place of $\phi_g(x)$.

A set $X$ endowed with a group action of $G$ is referred to as a *$G$-set*. That is, $X$ is a $G$-set if there exists a well-defined action $\cdot : G \times X \to X$ satisfying the identity and compatibility conditions above.

Another fundamental notion for our analysis is that of a map between $G$-sets that respects the group action. This leads to the definition of equivariance.

**Definition 26** (Equivariance). Let $X$ and $Y$ be $G$-sets. A function $f : X \to Y$ is said to be *G-equivariant* if, for all $g \in G$ and $x \in X$, the following condition holds:

$$f(g \cdot x) = g \cdot f(x).$$

### A.3   Group Representations and Equivariant Affine Transformations

Let $G$ be a group and $V$ be a vector space over a field $\mathbb{R}$. A $G$-action $\Phi : G \times V \to V$ on $V$ is *G-representation* if $\phi_g$ is linear for each $g$ in $G$. Or equivalently,

$$\phi : \begin{array}{c} G \to \mathrm{GL}(V) \\ g \mapsto \phi_g \end{array}$$

where $\mathrm{GL}(V)$ is the general linear group of $V$, consisting of all invertible linear transformations on $V$. We will usually identify the entire $\Phi : G \times V \to V$ action with $V$ itself and write $gv = \Phi(g, v)$.

Let $V$ and $W$ be two $G$-representations, we will indicate the set of equivariant linear maps between $V$ and $W$ as $\mathrm{Hom}_G(V, W)$ and as $\mathrm{Aff}_G(V, W)$ the set of equivariant affine maps. Note that $\mathrm{Hom}_G(V, W)$ is a vector space. Indeed, $0 \in \mathrm{Hom}_G(V, W)$ and for each $f, g \in \mathrm{Hom}_G(V, W)$ and each $\alpha, \beta \in \mathbb{R}$, $\alpha f + \beta g \in \mathrm{Hom}_G(V, W)$. The same is true for $\mathrm{Aff}_G(V, W)$.

Let $V$ be a $G$-representation, we define the set of invariant vectors $V^G = \{v \in V \mid gv = v \ \forall g \in G\}$.

### A.4   On Permutation Representations

**Definition 27.** Let $X$ be a finite set and let $G$ be a finite group acting on $X$. A *permutation representation* of $G$ is the linear action of $G$ on the space $\mathbb{R}^X$ defined by

$$g(e_x) = e_{g \cdot x} \quad \text{for all } g \in G, \ x \in X,$$

where $\{e_x\}_{x \in X}$ denotes the standard basis of $\mathbb{R}^X$.

**Proposition 28.** *Let $X$ and $Y$ be $G$-sets. Then, the following $G$-equivariant isomorphisms of representations hold:*

$$\mathbb{R}^{X \sqcup Y} \cong \mathbb{R}^X \oplus \mathbb{R}^Y \quad \text{and} \quad \mathbb{R}^{X \times Y} \cong \mathbb{R}^X \otimes \mathbb{R}^Y,$$

*where $X \sqcup Y$ denotes the disjoint union and $X \times Y$ the Cartesian product of the two sets.*

# B   On Commutative Algebra

For a general introduction to commutative algebra, we refer to Atiyah and MacDonald [57]. Here, we recall the notation necessary to prove Theorem 13, 14 and 15.

Let $\mathbb{R}[x_1, \ldots, x_n]$ denote the set of polynomials in the variables $x_1, \ldots, x_n$.

**Definition 29** (Ideal). An *ideal* $I$ of $\mathbb{R}[x_1, \ldots, x_n]$ is a subset such that, if $f \in I$, then $p \cdot f \in I$ for every $p \in \mathbb{R}[x_1, \ldots, x_n]$. If $X \subseteq \mathbb{R}^n$, we define

$$\mathcal{I}(X) = \{f \in \mathbb{R}[x_1, \ldots, x_n] \mid f(x) = 0 \ \forall x \in X\}.$$

**Definition 30** (Product of Ideals). Let $I, J \subseteq \mathbb{R}[x_1, \ldots, x_n]$ be ideals. Their product $I \cdot J$, or simply $IJ$, is the ideal defined by

$$IJ = \left\{ \sum_{k=1}^{r} f_k g_k \mid f_k \in I, \ g_k \in J, \ r \in \mathbb{N} \right\}.$$

**Definition 31** (Generators of an Ideal). Let $R = \mathbb{R}[x_1, \ldots, x_n]$ be the set of polynomial and let $f_1, \ldots, f_m \in R$. The *ideal generated* by $f_1, \ldots, f_m$ is the set

$$(f_1, \ldots, f_m) = \left\{ \sum_{i=1}^{m} h_i f_i \mid h_i \in R \right\}.$$

We say that $f_1, \ldots, f_m$ are *generators* of the ideal.

**Proposition 32.** *If $X$ is a linear subspace of $\mathbb{R}^n$ such that its orthogonal complement $X^\perp$ is spanned by vectors $v_1, \ldots, v_d$, then*

$$\mathcal{I}(X) = (v_1^\top \cdot x, \ldots, v_d^\top \cdot x).$$

*Proof.* Indeed,

$$\mathcal{I}(X) \supseteq (v_1^\top \cdot x, \ldots, v_d^\top \cdot x).$$

To prove the reverse inclusion, observe that—up to a change of coordinates—we may assume $v_i \cdot x = x_i$ for $i = 1, \ldots, d$. In this case, any polynomial $f(x) \in \mathcal{I}(X)$ can be written as

$$f(x) = a_1(x)x_1 + \cdots + a_d(x)x_d + b(x),$$

where $a_i(x) \in \mathbb{R}[x_1, \ldots, x_n]$ for each $i = 1, \ldots, d$, and $b(x)$ is a polynomial whose monomials do not involve the variables $x_1, \ldots, x_d$.

Now, since $f$ vanishes on $X = \{x \in \mathbb{R}^n : x_1 = \cdots = x_d = 0\}$, it must be that $b(x) = 0$ identically. Therefore, $f(x)$ lies in the ideal generated by $x_1, \ldots, x_d$, completing the proof. $\qquad\square$

*Remark* 33. The following are either standard results or direct consequences of the observations above:

- The intersection and the product of ideals are themselves ideals.

- $\mathcal{I}(X_1 \cup \cdots \cup X_\ell) = \mathcal{I}(X_1) \cap \cdots \cap \mathcal{I}(X_\ell)$.

- If $X_1, \ldots, X_\ell$ are linear subspaces of $\mathbb{R}^n$, then $\mathcal{I}(X_1) \cdots \mathcal{I}(X_\ell)$ is generated by polynomials of the form $(v_1^\top \cdot x) \cdots (v_\ell^\top \cdot x)$, where $v_1, \ldots, v_\ell$ are vectors respectively in $X_1^\perp, \ldots, X_\ell^\perp$.

## C   On Superpositions of Ridge Functions

In this section, we present results on the theory of superpositions of generalized ridge functions. A detailed exposition can be found in Pinkus [58].

**Definition 34** (Superpositions of Generalized Ridge Functions)**.** Given a linear map $\phi : \mathbb{R}^n \to \mathbb{R}^d$, a *generalized ridge functions* is an element in

$$\mathcal{M}(\phi) := \left\{ f \circ \phi \mid f \in \mathcal{C}(\mathbb{R}^d) \right\} \subseteq \mathcal{C}(\mathbb{R}^n).$$

Given $\Omega \subseteq \mathbb{R}^{d \times n}$, a *superposition of generalized ridge functions* is an element in

$$\mathcal{M}(\Omega) := \operatorname{Span} \left\{ f \circ \phi \mid f \in \mathcal{C}(\mathbb{R}^d), \, \phi \in \Omega \right\}.$$

If $\Omega$ is finite, say $\Omega = \{\phi_i\}_{i \in I}$, we may write

$$\mathcal{M}(\Omega) = \mathcal{M}(\phi_i)_{i \in I} := \left\{ x \mapsto \sum_{i \in I} f_i \circ \phi_i(x) \mid f_i \in \mathcal{C}(\mathbb{R}^d) \right\},$$

or simply write $\mathcal{M}(\phi_1, \ldots, \phi_l)$ when $\Omega = \{\phi_1, \ldots, \phi_l\}$.

To facilitate our exposition, we introduce the following auxiliary notation. Let $A \in \mathbb{R}^{d \times n}$ be matrix, and write it as

$$A := \begin{bmatrix} a_1 \\ \vdots \\ a_d \end{bmatrix},$$

where $a_i$s are the rows of $A$. Define

$$L(A) := \operatorname{Span} \{a_1, \ldots, a_d\}.$$

Let $\Omega \subseteq \mathbb{R}^{d \times n}$ be a finite set of matrices. Define

$$L(\Omega) := \bigcup_{A \in \Omega} L(A).$$

In the following, we will use the following fundamental result (see [58], p. 65).

**Theorem 35.** *Let $\Omega = \{A_1, \ldots, A_s\} \subseteq \mathbb{R}^{d \times n}$ be a finite set of matrices. Then*

$$\overline{\mathcal{M}(\Omega)} = \overline{\mathcal{M}(L(\Omega))} = \overline{\mathcal{M}(L(A_1) \cup \cdots \cup L(A_s))}.$$

We can characterize the previous sets using the following notions.

**Definition 36.** Given $\Omega \subseteq \mathbb{R}^n$, define the ideal of polynomials vanishing on $\Omega$ as

$$\mathcal{I}(\Omega) := \{p \in \mathbb{R}[x_1, \ldots, x_n] \mid p(x) = 0 \, \forall x \in \Omega\},$$

and then, define

$$\mathcal{C}(\Omega) := \{p \in \mathbb{R}[x_1, \ldots, x_n] \mid q(D)p = 0 \, \forall q \in \mathcal{I}(\Omega)\}.$$

**Theorem 37** (Theorem 6.9 of [58])**.** *In the topology of uniform convergence on compact subsets*

$$\overline{\mathcal{M}(\Omega)} = \overline{\mathcal{C}(\Omega)}.$$

We can compare the closure of spaces of superposition thanks to the following theorem.

**Theorem 38.** *Let $\Omega$ and $\Omega'$ be two subsets of $\mathbb{R}^n$ closed under scalar multiplication. If $\mathcal{C}(\Omega) \subsetneq \mathcal{C}(\Omega')$, then $\overline{\mathcal{C}(\Omega)} \subsetneq \overline{\mathcal{C}(\Omega')}$ in topology of uniform convergence on compact sets.*

*Proof.* If $\mathcal{C}(\Omega) \subsetneq \mathcal{C}(\Omega')$ then there exist $p' \in \mathcal{C}(\Omega')$ and $q \in \mathcal{I}(\Omega)$ such that

$$q'(D) \cdot p' = 0, \ \forall q' \in \mathcal{I}(\Omega')$$

and

$$q(D) \cdot p' \neq 0$$

for each $p \in \mathcal{C}(\Omega)$. Note that $q(D)$ is a continuous operator in the space of tempered distributions and $C(\Omega) \subseteq \ker q(D)$. Since $\ker q(D)$ is a closed subspace by Lemma 39, then $\overline{C(\Omega)} \subseteq \ker q(D)$ while $p' \notin \ker q(D)$, concluding the proof. $\square$

**Lemma 39.** *Let $(p_n)_{n \in \mathbb{N}}$ be a sequence of polynomials in $d$ variables, each of arbitrary degree, that converges uniformly on compact subsets to a polynomial $p$. Let $P(\partial_1, \ldots, \partial_d)$ be a linear differential operator with constant coefficients, that is, $P$ is a polynomial in $d$ variables. If*

$$P(\partial_1, \ldots, \partial_d) \, p_n = 0 \quad \text{for all } n \in \mathbb{N},$$

*then*

$$P(\partial_1, \ldots, \partial_d) \, p = 0.$$

*Proof.* Define:

$$\langle f, g \rangle := \int_{\mathbb{R}^n} f(x)g(x)dx.$$

Let $\phi$ be a smooth function with support on a compact $K$. We know:

$$\langle p_n, \phi \rangle \to \langle p, \phi \rangle,$$

for $n \to \infty$. Let $Q(\partial_1, \ldots, \partial_d)$ be the adjoint operator of $P(\partial_1, \ldots, \partial_d)$. This operator is still a linear differential operator when defined on smooth functions with compact support. In particular, $Q(\partial_1, \ldots, \partial_d)\phi$ is still a smooth function with support on $K$. Moreover,

$$\langle p_n, Q(\partial_1, \ldots, \partial_d)\phi \rangle = -\langle P(\partial_1, \ldots, \partial_d)p_n, \phi \rangle = 0 \tag{8}$$

for each $n$. Due to convergence on compacts and knowing that the support of $Q(\partial_1, \ldots, \partial_d)\phi$ is $K$, we obtain

$$\langle p_n, Q(\partial_1, \ldots, \partial_d)\phi \rangle \to \langle p, Q(\partial_1, \ldots, \partial_d)\phi \rangle, \tag{9}$$

for $n \to \infty$. Thanks to Eq. 8 e 9 we get:

$$\langle p, Q(\partial_1, \ldots, \partial_d)\phi \rangle = 0.$$

Finally,

$$\langle P(\partial_1, \ldots, \partial_d)p, \phi \rangle = -\langle p, Q(\partial_1, \ldots, \partial_d)\phi \rangle = 0.$$

Since $\phi$ is an arbitrary smooth function with compact support, we get

$$\langle P(\partial_1, \ldots, \partial_d)p, \phi \rangle = 0$$

for each $\phi$ with compact support. For the fundamental theorem of calculus of variations, $P(\partial_1, \ldots, \partial_d)p$ is identically zero. $\square$

# D   Proofs and Auxiliary Results

In this section we will concentrate on a particular subset of superpositions of ridge functions, namely, the symmetric ones.

**Definition 40** (Symmetric Superpositions)**.** Let $\phi_1, \dots, \phi_\ell : \mathbb{R}^n \to \mathbb{R}^d$ be linear maps. We define symmetric superpositions of ridge functions as follows:

$$\Delta(\phi_1, \dots, \phi_\ell) := \left\{ x \mapsto f \circ \phi_1(x) + \dots + f \circ \phi_\ell \mid f \in \mathcal{C}(\mathbb{R}^d) \right\}.$$

**Proposition 41.** *The family of functions approximated by $\mathcal{U}_\sigma(M, N)$ coincides with the class $\Delta(\phi_1, \dots, \phi_\ell)$, where $\phi_1, \dots, \phi_\ell$ are the basis maps associate to $M$.*

*Proof of Proposition 41.* In the general setting, write the linear parts of $M$ and $N$ respectively as $\lambda(M) = \mathrm{Span}\left\{ \phi^1, \dots, \phi^m \right\}$ and $\lambda(N) = \mathrm{Span}\left\{ x \mapsto \mathbb{1}^t \cdot x \right\}$. Elements in $M_h$ can be represented as affine maps $x \mapsto Bx + c$ where $B$ and $c$ have the following block representations

$$B = \begin{bmatrix} b_{1,1}\phi^1 + \dots + b_{1,m}\phi^m \\ \vdots \\ b_{h,1}\phi^1 + \dots + b_{h,m}\phi^m \end{bmatrix} \quad \text{and} \quad c = \begin{bmatrix} c_1 \, \mathbb{1} \\ \vdots \\ c_h \, \mathbb{1} \end{bmatrix}.$$

While elements in $_hN$ can be represented as affine maps $x \mapsto Ax + d$ where $d \in \mathbb{R}$ and

$$A = \begin{bmatrix} a_1 \, \mathbb{1}^t \\ \vdots \\ a_h \, \mathbb{1}^t \end{bmatrix}.$$

Denote by $\phi_i^j$ the projection of the $i$-th component of the function $\phi^j$. We can write elements $\eta \in \mathcal{N}_\sigma(M_{h,h}\, N)$ as

$$\eta(x) = A\sigma(Bx + c) = \sum_{j=1}^{h} a_j \sum_{i \in Y} \sigma \left( \sum_{t=1}^{m} b_{j,t}\phi_i^t(x) + c_j \right)$$

for some $a_i, b_{j,t}, c_j \in \mathbb{R}$. But note that

$$\eta(x) = \sum_{j=1}^{h} a_j \sum_{i \in Y} \sigma \left( \sum_{t=1}^{m} b_{j,t}\phi_i^t(x) + c_j \right) = \tag{10}$$

$$\sum_{i \in Y} \sum_{j=1}^{h} a_j \sigma \left( \sum_{t=1}^{m} b_{j,t}\phi_i^t(x) + c_j \right) = \sum_{i \in Y} \zeta(\phi_i^1(x), \dots, \phi_i^m(x)) \tag{11}$$

where

$$\zeta(y_1, \dots, y_l) := \sum_{j=1}^{h} a_j \sigma \left( \sum_{t=1}^{m} b_{j,t}y_t + c_j \right)$$

is a standard multilayer perceptron in $\mathcal{N}_\sigma(\mathbb{R}^l, \mathbb{R}^h, \mathbb{R})$. Since, the the multilayer perceptron is universal, thanks to (10) we can approximate any superposition in $\Delta(\phi_1, \dots, \phi_l)$. Thus, we have

$$\overline{\Delta(\phi_1, \dots, \phi_l)} \subseteq \mathcal{U}_\sigma(M, N).$$

On the other hand, by (10),

$$\bigcup_{h \in \mathbb{N}} \mathcal{N}_\sigma(M_{h,h}\, N) \subseteq \Delta(\phi_1, \dots, \phi_l).$$

It follows that their closures coincide, which concludes the proof. $\qquad\square$

Let $M$ be a vector space of affine maps such that $\lambda(M) = \mathrm{Span}\left\{ \phi^1, \dots, \phi^m \right\}$, and let $N$ be the set of invariant affine maps. Denote $\rho = \rho(\mathcal{N}_\sigma(M, N))$. We denote by $\{\{x_1, \dots, x_n\}\}$ the multiset of elements $x_1, \dots, x_n$. We have the following proposition.

**Proposition 42.** *With the notation defined above, we have $(x,y) \in \rho(\mathcal{N}_\sigma(M,N)) = \rho(\mathcal{U}_\sigma(M,N))$ if and only if*

$$\{\{\phi_1(x),\dots,\phi_\ell(x)\}\} = \{\{\phi_1(y),\dots,\phi_\ell(y)\}\},$$

*where we identify $Y$ with $[\ell]$, which inherits its $G$-set structure from $Y$, and the maps $\phi_i$ are those defined in* (6).

*Proof.* By the combination of Proposition 41 and Theorem 8 in [18], we have $\rho(\mathcal{N}_\sigma(M,N)) = \rho(\mathcal{U}_\sigma(M,N)) = \rho(\Delta(\phi_1,\dots,\phi_\ell))$. Thus, it suffices to verify this property for $\Delta(\phi_1,\dots,\phi_\ell)$. Note that if $x$ and $y$ satisfy

$$\{\{\phi_1(x),\dots,\phi_\ell(x)\}\} = \{\{\phi_1(y),\dots,\phi_\ell(y)\}\},$$

then, for each $F \in \Delta(\phi_1,\dots,\phi_\ell)$, we have

$$F(x) = f \circ \phi_1(x) + \cdots + f \circ \phi_\ell(x) = f \circ \phi_1(y) + \cdots + f \circ \phi_\ell(y) = F(y).$$

On the other hand, if

$$\{\{\phi_1(x),\dots,\phi_\ell(x)\}\} \neq \{\{\phi_1(y),\dots,\phi_\ell(y)\}\},$$

then we have two possibilities: either there exists an $i$ such that $\phi_i(x) \neq \phi_i(y)$, or there exists a value $\gamma$ such that the number of indices $i$ with $\phi_i(x) = \gamma$ (denoted $s$) differs from the number of indices $i$ with $\phi_i(y) = \gamma$ (denoted $t$). In the first case, we can choose an interpolating function $f$ that does not vanish at $\phi_i(x)$ and is zero on the other values in consideration. In this case,

$$F(x) = f \circ \phi_1(x) + \cdots + f \circ \phi_\ell(x) \neq 0 = f \circ \phi_1(y) + \cdots + f \circ \phi_\ell(y) = F(y).$$

In the other case, we can similarly chose a function $f$ nonzero on $\gamma$ and zero on all the other values in consideration. In this case,

$$F(x) = f \circ \phi_1(x) + \cdots + f \circ \phi_\ell(x) = sf(\gamma) \neq tf(\gamma) = f \circ \phi_1(y) + \cdots + f \circ \phi_\ell(y) = F(y).$$

This concludes the proof. $\qquad\square$

Proposition 9 follows directly from Proposition 42.

*Proof of Proposition 9.* Note that, by Theorem 6, $\rho(\mathcal{U}_\sigma(C^1,I)) = \mathcal{C}_{S_n}(\mathbb{R}^n)$ and thus has maximal separation power in the context of permutation invariance; that is, it separates two points if and only if they lie in the same $S_n$-orbit. Note that the basis maps associated to $C^1$ are $e_1^\top,\dots,e_n^\top$. Hence, by Proposition 42, $(x,y) \in \rho(\mathcal{U}_\sigma(C^1,I))$ if and only if $\{\{x_1,\dots,x_n\}\} = \{\{y_1,\dots,y_n\}\}$. This holds if and only if $x$ and $y$ lie in the same $S_n$-orbit. Thus, $\mathcal{U}_\sigma(C^1,I)$ also has maximal separation power, and hence

$$\rho(\mathcal{U}_\sigma(\mathbb{R}^n,\mathbb{R}^G,\mathbb{R})) = \rho(\mathcal{U}_\sigma(C^1,I)).$$

Since

$$\mathcal{U}_\sigma(C^1,I) \subseteq \mathcal{U}_\sigma(\mathbb{R}^n,\mathbb{R}^n,\mathbb{R}) \subseteq \mathcal{U}_\sigma(\mathbb{R}^n,\mathbb{R}^G,\mathbb{R}),$$

it follows that

$$\rho(\mathcal{U}_\sigma(\mathbb{R}^n,\mathbb{R}^G,\mathbb{R})) \subseteq \rho(\mathcal{U}_\sigma(\mathbb{R}^n,\mathbb{R}^n,\mathbb{R})) \subseteq \rho(\mathcal{U}_\sigma(C^1,I)).$$

Therefore, all inclusions must be equalities. $\qquad\square$

*Proof of Theorem 13.* Note that

$$\mathcal{M}(\phi_1,\dots,\phi_\ell) = \mathcal{M}(\psi_1,\dots,\psi_\ell)$$

for basis maps $\phi_1,\dots,\phi_\ell$ and $\psi_1,\dots,\psi_\ell$ such that

$$\mathrm{Span}\{\phi^1,\dots,\phi^m\} = \mathrm{Span}\{\psi^1,\dots,\psi^m\},$$

since

$$L(\phi_i) = L(\psi_i)$$

for each $i = 1,\dots,\ell$. In particular,

$$\mathcal{M}(\phi_1,\dots,\phi_\ell)^G = \mathcal{M}(\psi_1,\dots,\psi_\ell)^G.$$

Thus, it suffices to restrict to a specific basis $\phi^1,\dots,\phi^m$ chosen as follows.

Consider the decomposition

$$\operatorname{Hom}_G(\mathbb{R}^X, \mathbb{R}^Y) \cong \operatorname{Hom}_G(\mathbb{R}^X, \mathbb{R}^{Y_1}) \oplus \cdots \oplus \operatorname{Hom}_G(\mathbb{R}^X, \mathbb{R}^{Y_r}),$$

and construct a basis of $\lambda(M)$ by choosing, for each $i = 1, \ldots, r$, a basis of $\operatorname{Hom}_G(\mathbb{R}^X, \mathbb{R}^{Y_i})$, and embedding it in $\operatorname{Hom}_G(\mathbb{R}^X, \mathbb{R}^Y)$ via the canonical inclusion induced by the direct sum decomposition. In particular, there exists a partition

$$\mathcal{I}_1 \sqcup \cdots \sqcup \mathcal{I}_r = [m]$$

such that

$$\operatorname{Span}\{\phi^j\}_{j \in \mathcal{I}_i} = \operatorname{Hom}_G(\mathbb{R}^X, \mathbb{R}^{Y_i})$$

for each $i = 1, \ldots, r$, where each $\operatorname{Hom}_G(\mathbb{R}^X, \mathbb{R}^{Y_i})$ is viewed as embedded in $\operatorname{Hom}_G(\mathbb{R}^X, \mathbb{R}^Y)$. Equivalently, for each $j \in \mathcal{I}_i$ there exists $\psi^j \in \operatorname{Hom}_G(\mathbb{R}^X, \mathbb{R}^{Y_i})$ such that

$$\phi^j(x) = (0, \ldots, 0, \underbrace{\psi^j(x)}_{i\text{-th block}}, 0, \ldots, 0). \tag{12}$$

With this notation in place, we now prove that

$$\mathcal{M}(\phi_1, \ldots, \phi_\ell)^G = \Delta(\phi_1, \ldots, \phi_\ell).$$

Let $\mathcal{R} : \mathcal{C}(\mathbb{R}^n) \to \mathcal{C}(\mathbb{R}^n)^G$ be the Reynolds operator, namely

$$\mathcal{R}(F)(x) := \frac{1}{|G|} \sum_{g \in G} F(gx).$$

For each $F \in \mathcal{M}(\phi_1, \ldots, \phi_\ell)$, write $F(x) = \sum_{k=1}^\ell f_k \circ \phi_k(x)$ for some continuous $f_1, \ldots, f_\ell$. Moreover, since $\phi^1, \ldots, \phi^m$ are $G$-equivariant, the induced $G$-action permutes the corresponding family of coordinate maps. More precisely, for each $g \in G$ and each index $k$ we have

$$\phi_k(gx) = \phi_{g^{-1} \cdot k}(x) \qquad \text{for all } x.$$

Indeed, using equivariance of each $\phi^j$,

$$\left( \phi_k^1(gx), \ldots, \phi_k^m(gx) \right) = \left( (g\phi^1)_k(x), \ldots, (g\phi^m)_k(x) \right)$$
$$= \left( \phi_{g^{-1} \cdot k}^1(x), \ldots, \phi_{g^{-1} \cdot k}^m(x) \right),$$

for each $k = 1, \ldots, \ell$. Then

$$\mathcal{R}(F)(x) = \frac{1}{|G|} \sum_{g \in G} F(gx) = \frac{1}{|G|} \sum_{g \in G} \sum_{k=1}^\ell f_k \circ \phi_k(gx).$$

$$= \frac{1}{|G|} \sum_{g \in G} \sum_{k=1}^\ell f_k \circ \phi_{g \cdot k}(x).$$

Next, group together indices belonging to the same block. For each $i = 1, \ldots, r$, let $\pi_i : \mathbb{R}^m \to \mathbb{R}^{|\mathcal{I}_i|}$ be the coordinate projection induced by the partition $[m] = \mathcal{I}_1 \sqcup \cdots \sqcup \mathcal{I}_r$. That is,

$$\pi_i(z_1, \ldots, z_m) := (z_j)_{j \in \mathcal{I}_i}.$$

In particular, for each $x$ we have

$$\pi_i\left( \phi_h^1(x), \ldots, \phi_h^m(x) \right) = \left( \phi_h^j(x) \right)_{j \in \mathcal{I}_i} = \left( \psi_h^j(x) \right)_{j \in \mathcal{I}_i}.$$

Define

$$F_i := \frac{1}{|G|} \sum_{j \in \mathcal{I}_i} f_j \circ \pi_i, \qquad \text{and} \qquad H(y_1, \ldots, y_r) := \sum_{i=1}^r F_i(y_i).$$

Then, using the block form in (12),

$$\mathcal{R}(F) = \frac{1}{|G|} \sum_{g \in G} \sum_{k=1}^{\ell} f_k \circ \phi_{g \cdot k}$$

$$= \sum_{i=1}^{r} \sum_{j \in \mathcal{I}_i} F_i \circ \psi_j$$

$$= \sum_{i=1}^{r} \sum_{j \in \mathcal{I}_i} H \circ \phi_j$$

$$= \sum_{j=1}^{\ell} H \circ \phi_j.$$

The right-hand side is a sum of terms of the form $\sum_{j=1}^{\ell} H \circ \phi_j$, hence it belongs to $\Delta(\phi_1, \ldots, \phi_\ell)$. Moreover, the preceding regrouping shows that $\mathcal{R}(F)$ can be written in this form, so $\mathcal{R}(F) \in \Delta(\phi_1, \ldots, \phi_\ell)$. Therefore,

$$\mathcal{M}(\phi_1, \ldots, \phi_\ell)^G \subseteq \Delta(\phi_1, \ldots, \phi_\ell).$$

Conversely, since $\Delta(\phi_1, \ldots, \phi_\ell)$ is $G$-invariant by construction, we have

$$\Delta(\phi_1, \ldots, \phi_\ell)^G = \Delta(\phi_1, \ldots, \phi_\ell) \subseteq \mathcal{M}(\phi_1, \ldots, \phi_\ell).$$

Combining the two inclusions yields

$$\mathcal{M}(\phi_1, \ldots, \phi_\ell)^G = \Delta(\phi_1, \ldots, \phi_\ell).$$

Finally, if $f$ is an invariant function, then by Theorem 37 and Theorem 41 we have

$$f \in \mathcal{U}_\sigma(M, N) \iff f \in \overline{\Delta(\phi_1, \ldots, \phi_\ell)}.$$
$$\iff f \in \overline{\mathcal{M}(\phi_1, \ldots, \phi_\ell)} \iff f \in \overline{\mathcal{C}(\phi_1, \ldots, \phi_\ell)}.$$
$$\iff P(\partial_1, \ldots, \partial_n)f = 0 \quad \text{for each } P \in \mathcal{I}\big(L(\phi_1) \cup \cdots \cup L(\phi_\ell)\big).$$

This concludes the proof. $\qquad\square$

*Proof of Theorem 14.* The final part of the proof of Theorem 13 implies that if $f \in \mathcal{U}_\sigma(M, N)$, then for any $P \in \mathcal{I}(L(\phi_1) \cup \cdots \cup L(\phi_\ell))$, $P(\partial_1, \ldots, \partial_n)f = 0$. By Remark 33, we know

$$\mathcal{I}(L(\phi_1) \cup \cdots \cup L(\phi_\ell)) = \mathcal{I}(L(\phi_1)) \cap \cdots \cap \mathcal{I}(L(\phi_\ell)) \supseteq \mathcal{I}(L(\phi_1)) \cdots \mathcal{I}(L(\phi_\ell)).$$

For any $\alpha = 1, \ldots, \ell$ and arbitrary $c_\alpha \in \ker \phi_\alpha^\top$, note that for

$$c_\alpha^\top x \in \mathcal{I}(L(\phi_\alpha)).$$

Hence,

$$(c_1^\top x) \cdots (c_\ell^\top x) \in \mathcal{I}(L(\phi_1)) \cdots \mathcal{I}(L(\phi_\ell)).$$

Whose associated differential operator can be written as $D_{c_1} \cdots D_{c_\ell}$. Therefore,

$$D_{c_1} \cdots D_{c_\ell} f = 0,$$

concluding the proof. $\qquad\square$

*Proof of Theorem 15.* By Proposition 42, separation-constrained universality is equivalent to the ability to approximate any function of the form $F(\phi_1, \ldots, \phi_\ell)$, where $F$ is continuous and $S_\ell$-invariant.

Recall that the basis maps are defined as

$$\phi_i = (\phi_i^1, \ldots, \phi_i^m).$$

Let $W = \mathbb{R}^Y$ for some finite $G$-set $Y$. Since $M = \text{Aff}_G(V, \mathbb{R}^Y)$, we can, for a suitable choice of basis, select elements $\alpha_i \in Y$ such that $\phi_i^1 = e_{\alpha_i}^\top$ for each $i = 1, \ldots, \ell$.

In particular, the function
$$F : x \mapsto G(e_{\alpha_1}^\top x, \ldots, e_{\alpha_\ell}^\top x),$$
for some $G : \mathbb{R}^\ell \to \mathbb{R}$, is one that should be approximable under separation constraints.

Specifically, we define $G$ as the symmetrization of the monomial
$$M(x_1, \ldots, x_\ell) = x_1^{a_1} \cdots x_\ell^{a_\ell},$$
that is,
$$G(x_1, \ldots, x_\ell) = \sum_{\sigma \in S_\ell} M(x_{\sigma(1)}, \ldots, x_{\sigma(\ell)}).$$

Now, observe that if
$$D_{c_1} \cdots D_{c_\ell} M \neq 0,$$
then
$$D_{c_1} \cdots D_{c_\ell} G \neq 0$$
for any choice of $c_i \in \ker \phi_i$ for some $i = 1, \ldots, \ell$. Therefore, $F$ cannot be approximated by $\bigcup_{h \in \mathbb{N}} \mathcal{N}(M_{h,h} N)$.

This follows because the differential operator $D_{c_1} \cdots D_{c_\ell}$ reduces the degree of each monomial in $G$ by at most $\ell$. Thanks to the hypothesis $a_i + \ell < a_{i+1}$ for each $i = 1, \ldots, \ell$, and $a_1 > \ell$, all resulting monomials in $D_{c_1} \cdots D_{c_\ell} G$ have distinct multidegrees. In particular, $D_{c_1} \cdots D_{c_\ell} M$, being one of these monomials and being nonzero, implies that $D_{c_1} \cdots D_{c_\ell} G$ is itself nontrivial.

This proves that if $D_{c_1} \cdots D_{c_\ell} M \neq 0$, then the function $F$ cannot be approximated by $\bigcup_{h \in \mathbb{N}} \mathcal{N}(M_{h,h} N)$.

By direct computation, the coefficients of the monomials of multidegree $(a_1 - s_1, \ldots, a_\ell - s_\ell)$ in $D_{c_1} \cdots D_{c_\ell} M$ are given by
$$\sum_{\sigma \in S_\ell} \frac{a_{i_1}!}{s_{i_1}!} \cdots \frac{a_{i_r}!}{s_{i_r}!} (c_{\sigma(1),1} \cdots c_{\sigma(s_1),1}) \cdot (c_{\sigma(s_1+1),2} \cdots c_{\sigma(s_1+s_2),2}) \cdots (c_{\sigma(\ell-s_\ell),\ell} \cdots c_{\ell,\ell}).$$

where $s_1, \ldots, s_\ell \in \{0, \ldots, \ell\}$, $s_1 + \cdots + s_\ell = \ell$ and $i_1, \ldots, i_r$ are the indices such that $s_{i_j} \neq 0$.

If at least one of these coefficients is nonzero, then $D_{c_1} \cdots D_{c_\ell} F$ is nontrivial and thus cannot be approximated by $\bigcup_{h \in \mathbb{N}} \mathcal{N}(M_{h,h} N)$. $\qquad\square$

*Proof of Theorem 18.* Define $V$, $W$, and $\iota : V \to W$ as in Corollary 44, which states that
$$\mathcal{N}(V, \mathbb{R}^{G/H} \otimes \mathbb{R}^h, Z) = \iota^* \mathcal{N}(W, \mathbb{R}^{G/H} \otimes \mathbb{R}^h, Z)$$
for each $h \in \mathbb{N}$.

Since $H$ is normal in $G$, the quotient $G/H$ is a group and the action of $H$ on $W$ is trivial, $W$ is a $G/H$-representation, and we have the identification $\mathcal{C}_G(W, Z) = \mathcal{C}_{G/H}(W, Z)$.

From Ravanbakhsh [5], it is known that shallow equivariant neural networks with the regular representation as input are universal approximators. In this case,
$$\bigcup_{h \in \mathbb{N}} \mathcal{N}(W, \mathbb{R}^{G/H} \otimes \mathbb{R}^h, Z)$$
is universal in $\mathcal{C}_G(W, Z) = \mathcal{C}_{G/H}(W, Z)$.

Furthermore, the pullback map $\iota^* : \mathcal{C}(V, Z) \to \mathcal{C}(W, Z)$ is a continuous linear operator. Hence,
$$\overline{\bigcup_{h \in \mathbb{N}} \mathcal{N}(V, \mathbb{R}^{G/H} \otimes \mathbb{R}^h, Z)} = \overline{\bigcup_{h \in \mathbb{N}} \iota^* \mathcal{N}(W, \mathbb{R}^{G/H} \otimes \mathbb{R}^h, Z)}$$
$$= \iota^* \left( \overline{\bigcup_{h \in \mathbb{N}} \mathcal{N}(W, \mathbb{R}^{G/H} \otimes \mathbb{R}^h, Z)} \right)$$
$$= \iota^* \left( \overline{\bigcup_{h \in \mathbb{N}} \mathcal{N}(W, \mathbb{R}^{G/H} \otimes \mathbb{R}^h, Z)} \right)$$
$$= \iota^* \left( \mathcal{C}_{G/H}(W, Z) \right) = \iota^* \left( \mathcal{C}_G(W, Z) \right).$$

Therefore, the left-hand side is equivariant-universal as well. Finally, observe that $\iota^*(\mathcal{C}_G(W, Z))$ is an algebra of functions containing the constants, so it is separation-constrained universal by the Stone–Weierstrass theorem. $\qquad\square$

**Lemma 43.** *Let $H$ be normal subgroup of $G$ and $K$ an arbitrary subgroup of $G$. Consider the standard immersion map*

$$\iota : \mathbb{R}^{G/HK} \to \mathbb{R}^{G/K}$$

*as the standard injection induced by the subgroup inclusion $K < KH$. We define the pullback map*

$$\iota^* : \begin{array}{c} \mathcal{C}(\mathbb{R}^{G/K}, Z) \to \mathcal{C}(\mathbb{R}^{G/HK}, Z) \\ f \mapsto f \circ \iota \end{array}$$

*for any $G$-representation $Z$.*

*Proof.* Note that $\iota^* \operatorname{Hom}_G(\mathbb{R}^{G/K}, \mathbb{R}^{G/H}) \subseteq \operatorname{Hom}_G(\mathbb{R}^{G/HK}, \mathbb{R}^{G/H})$, since $\iota^*$ is linear and preserves equivariance. Moreover, since $\iota$ is injective, the induced map $\iota^*$ is surjective.

Now, assume that $H$ is normal. Then,

$$\dim \operatorname{Hom}_G(\mathbb{R}^{G/K}, \mathbb{R}^{G/H}) = |H\backslash G/K| = |H\backslash G/HK| = \dim \operatorname{Hom}_G(\mathbb{R}^{G/HK}, \mathbb{R}^{G/H}).$$

This equality of dimensions, together with the inclusion and surjectivity above, implies that $\iota^*$ is an isomorphism of vector spaces. In particular,

$$\iota^* \operatorname{Hom}_G(\mathbb{R}^{G/K}, \mathbb{R}^{G/H}) = \operatorname{Hom}_G(\mathbb{R}^{G/HK}, \mathbb{R}^{G/H}).$$

$\qquad\square$

**Corollary 44.** *Let $V = \mathbb{R}^{G/K_1} \oplus \cdots \oplus \mathbb{R}^{G/K_d}$ and define $W = \mathbb{R}^{G/K_1 H} \oplus \cdots \oplus \mathbb{R}^{G/K_d H}$. Consider the standard immersion map $\iota : W \to V$ as the standard injection defined component by component and induced by the subgroup inclusion $K_i < K_i H$ for $i = 1, \ldots, d$. We define the pullback map*

$$\iota^* : \begin{array}{c} \mathcal{C}(V, Z) \to \mathcal{C}(W, Z) \\ f \mapsto f \circ \iota \end{array}$$

*for any $G$-representation $Z$. Then*

$$\mathcal{N}(V, \mathbb{R}^{G/H} \otimes \mathbb{R}^h, Z) = \iota^* \mathcal{N}(W, \mathbb{R}^{G/H} \otimes \mathbb{R}^h, Z),$$

*for any $G$-representation $Z$.*

*Proof.* By the properties of representation homomorphisms under direct sums, we have

$$\operatorname{Hom}_G(V, \mathbb{R}^{G/H} \otimes \mathbb{R}^h) = \operatorname{Hom}_G \left( \mathbb{R}^{G/K_1} \oplus \cdots \oplus \mathbb{R}^{G/K_d}, \mathbb{R}^{G/H} \otimes \mathbb{R}^h \right)$$

$$= \bigoplus_{i=1}^d \operatorname{Hom}_G(\mathbb{R}^{G/K_i}, \mathbb{R}^{G/H})^{\oplus h}.$$

By the definition of $\iota$ and Lemma 43, it follows that

$$\iota^* \operatorname{Hom}_G(V, \mathbb{R}^{G/H} \otimes \mathbb{R}^h) = \operatorname{Hom}_G(W, \mathbb{R}^{G/H} \otimes \mathbb{R}^h)$$

for each $h \in \mathbb{N}$. Consequently,

$$\iota^* \operatorname{Aff}_G(V, \mathbb{R}^{G/H} \otimes \mathbb{R}^h) = \operatorname{Aff}_G(W, \mathbb{R}^{G/H} \otimes \mathbb{R}^h)$$

for every $h \in \mathbb{N}$ as well.

Therefore, for any $G$-representation $Z$, we obtain

$$\mathcal{N}(V, \mathbb{R}^{G/H} \otimes \mathbb{R}^h, Z) = \iota^* \mathcal{N}(W, \mathbb{R}^{G/H} \otimes \mathbb{R}^h, Z),$$

since $\iota$ is precomposed with the input in the first layer. $\qquad\square$

