# OpenReview forum: "On Universality Classes of Equivariant Networks"
_NeurIPS.cc/2025/Conference — NeurIPS 2025 spotlight_

### Official Review · Reviewer_ETMU · 2025-06-11

**Clarity:** 2
**Significance:** 3
**Originality:** 3
**Rating:** 4
**Confidence:** 2

**Summary:**

This paper provides a theoretical investigation into the approximation capabilities (universality) of equivariant neural networks, arguing that this property should be considered distinct from the more commonly studied separation power. The authors develop a framework for analyzing the universality classes of shallow invariant networks by connecting them to the theory of generalized ridge functions. This framework characterizes the functions a network can approximate via a set of differential equations they must satisfy.

The main contribution is a formal characterization of the universality classes for shallow invariant networks, and a demonstration for concrete examples (shallow CNNs and PointNets), that models with identical separation power can have different approximation capabilities. This directly shows that separation power is not a complete measure of expressivity. Further, the authors generalize prior universality results and show that shallow networks can be universal if the symmetry group possesses suitable normal subgroups, a condition that highlights structural limitations for important cases like the permutation group.

**Questions:**

1. The results clearly highlight the limitations of shallow permutation-equivariant networks. You contrast this with prior work showing depth-3 PointNets are universal. Could you speculate on how your analytical framework could be extended to deeper networks? Would a recursive application of the differential operator analysis for each layer be sufficient, or would a fundamentally different approach be necessary?
2. Your results suggest that for groups like $S_n$, standard shallow equivariant layers are not universal. What are the practical takeaways for an ML engineer designing a permutation-equivariant model? Should they always build deep models? Or does your theory suggest alternative designs for the hidden layers that could achieve universality even with limited depth?
3. Instead of strictly enforcing symmetry by a particular architectural design, an alternative approach is to use an unconstrained architecture and learn the symmetries of interest from (augmented) data. Assuming such a model has learned the symmetries sufficiently well (i.e. it is approximately equivariant), would it have similar limitations? Phrased differently, is it possible to derive a sort of "tradeoff" between achievable approximation accuracy and equivariance for a (shallow) unconstrained model?

**Ethical Concerns:**

["NO or VERY MINOR ethics concerns only"]

**Final Justification:**

The authors acknowledge the weaknesses I pointed out, but there are no immediately obvious ways to address all of them. I think this is fine, as this contribution is an important first step. The work would of course be more impactful with extended results (which is why I keep my rating at 4), but I think it is legitimate to leave them for future work.

**Limitations:**

yes

**Quality:**

3

**Strengths And Weaknesses:**

**Strengths**

1. The paper's central message (separation power does not fully capture expressivity) is a nice conceptual contribution to the field. Much of the theoretical work on GNNs and other equivariant models has focused on separation, often using it as a proxy for overall expressivity. This paper convincingly demonstrates the limitations of that view.
2. The authors build a rigorous mathematical framework leveraging sophisticated tools from approximation theory (specifically, the theory of ridge functions and their characterization via differential operators). The main theorems are precise, and the proofs provided in the appendix appear sound to me (although I did not check everything carefully).

**Weaknesses**

1. The analysis is limited to shallow networks. While this is a necessary starting point for a deep theoretical analysis, it remains a key limitation. The authors point out that depth can be crucial for achieving universality in practice, but how exactly the interplay between depth, separation, and approximation power unfolds in deeper models remains a major open question.
2. The main characterization theorem (Theorem 13) and its corollaries are elegant but highly technical. For a practitioner wanting to analyze a new architecture, verifying the required conditions could be very challenging. The primary value of the theory may lie more in the conceptual insights it provides than as a plug-and-play analysis tool.
3. The fact that the paper is highly technical and dense in theory makes it comparatively hard to read. Given the topic, this is unavoidable to some degree, but I think the paper could benefit from illustrations that allow readers to develop an intuition about relevant concepts.

---

> ### Author Rebuttal · Authors · 2025-07-29
>
> We appreciate the reviewer’s thoughtful assessment and the recognition of the theoretical contribution and broader relevance. The questions and suggestions raised are addressed point by point in what follows.
>
> > The analysis is limited to shallow networks. While this is a necessary starting point for a deep theoretical analysis, it remains a key limitation. The authors point out that depth can be crucial for achieving universality in practice, but how exactly the interplay between depth, separation, and approximation power unfolds in deeper models remains a major open question.
>
> We agree that restricting the analysis to shallow networks is a limitation. However, we would like to emphasize that even within this setting, our results already answer the central question: *“Is separation a complete proxy for approximation?”* The negative examples we provide show that it is not, which is the main takeaway of the paper. With these results in place, we are now actively working on extending the analysis to deeper architectures.
>
> > The main characterization theorem (Theorem 13) and its corollaries are elegant but highly technical. For a practitioner wanting to analyze a new architecture, verifying the required conditions could be very challenging. The primary value of the theory may lie more in the conceptual insights it provides than as a plug-and-play analysis tool.
>
> We truly appreciate the reviewer’s remark on the elegance of the results and we acknowledge that the paper may resonate more strongly with researchers than with practitioners. Rather than a limitation, we see this as a defining feature of theory, whose purpose is to clarify concepts and provide the foundations of further investigation.
> In this respect, we believe that the conceptual insights provided here can ultimately benefit both research and practice, even without offering direct plug-and-play tools.
>
> > The fact that the paper is highly technical and dense in theory makes it comparatively hard to read. Given the topic, this is unavoidable to some degree, but I think the paper could benefit from illustrations that allow readers to develop an intuition about relevant concepts.
>
> We share the reviewer's concerns but, unfortunately, proving these statements necessarily required introducing some theoretical machinery. However, we will eventually use the additional page to include some further commentary to motivate and explain the results. We will add illustrations to improve readability and take advantage of the additional page to do so, as suggested by Reviewer Wpwm. Furthermore, we plan to add proof hints and additional small examples to better guide the reader through the technical details.
>
> > The results clearly highlight the limitations of shallow permutation-equivariant networks. You contrast this with prior work showing depth-3 PointNets are universal. Could you speculate on how your analytical framework could be extended to deeper networks? Would a recursive application of the differential operator analysis for each layer be sufficient, or would a fundamentally different approach be necessary?
>
> We are currently investigating a recursive approach. However, we are already encountering significant complications even in the three-layer case. In particular, it is not yet clear how the associated differential operators behave under composition, as the standard chain rule becomes substantially more intricate in this setting. For now, we are focusing on the three-layer case as a natural first step toward a recursive strategy, while remaining open to alternative techniques should this direction prove unfruitful. Since these are preliminary efforts which do not have yet yielded concrete results, we do not plan to mention in the paper.
>
> > Your results suggest that for groups like $S_n$, standard shallow equivariant layers are not universal. What are the practical takeaways for an ML engineer designing a permutation-equivariant model? Should they always build deep models? Or does your theory suggest alternative designs for the hidden layers that could achieve universality even with limited depth?
>
> As a simple *heuristics*, we would suggest practitioners to rely on depth as the easiest and most computationally efficient way to improve approximation power based on [3], which shows that 3-layer PointNets are universal, and [2], that shows that depth may increase separation power.
> However, it is worth to note that this suggestion remains a heuristics since, in our knowledge, there exists no work characterizing the role of depth in separation-contrained approximation.
>
> That said, we are aware that in popular equivariant models, such as graph neural networks, increasing depth can harm learning due to oversmoothing and oversquashing. A practitioner encountering these phenomena may need to explore alternative hidden representations and here our contribution is a *theoretically-grounded* guide in the choice of a more expressive model, depending on the representations involved. In particular, Theorem 18 shows that representations coming from normal subgroups achieve separation-constrained universality, while Proposition 16 provides examples of subgroups or layers that do not. To assess expressivity for any particular case not covered in the previous examples, it is necessary to directly apply Theorem 13.
>
> > Instead of strictly enforcing symmetry by a particular architectural design, an alternative approach is to use an unconstrained architecture and learn the symmetries of interest from (augmented) data. Assuming such a model has learned the symmetries sufficiently well (i.e. it is approximately equivariant), would it have similar limitations? Phrased differently, is it possible to derive a sort of "tradeoff" between achievable approximation accuracy and equivariance for a (shallow) unconstrained model?
>
> This is a highly relevant question whose answer is still the object of active research by the community. We believe that the next points to be relevant directions toward an answer:
> - A standard MLP can approximate functions such as multiplication arbitrarily well, whereas shallow PointNets cannot achieve this (as shown in Proposition 16). However, PointNets may still approximate such functions on a restricted domain up to a prescribed approximation error. This can be interpreted as a trade-off between equivariance, parameter efficiency, and universality; a direction we are actively investigating.
> - We are also currently validating results for MLPs that can exactly represent an equivariant network. Relaxing this setting to allow approximate representation is the next step we plan to investigate. In particular, we are exploring this direction in terms of the trade-off between parameter efficiency and approximation power.
> - Finally, it is worth noting that learning dynamics play a crucial role in the broader learning process, beyond approximation alone. Equivariant constraints may indeed act as a barrier to the learning dynamics, as highlighted in [1].
>
> ---
>
> [1] YuQing Xie, Tess Smidt, A Tale of Two Symmetries: Exploring the Loss Landscape of Equivariant Models, 2025
>
> [2] Pacini et al. Separation Power of Equivariant Neural Networks, 2025
>
> [3] N. Segol et al., On Universal Equivariant Set Networks, 2020

---

> > ### Comment · Reviewer_ETMU · 2025-08-01
> >
> > I thank the authors for their detailed and honest response. I fully agree with the authors that their theoretical contributions are interesting and important, even if perhaps not immediately useful to all practitioners. Nonetheless, I believe it very important to make the insights as accessible as possible (there are obviously limits to this!), and I am glad to hear that the authors strive to do so in the camera-ready version of the manuscript.
> >
> > I am looking forward to potential follow-ups of this work that include results on deeper model architectures (I fully acknowledge that this is very difficult and the current results on shallow architectures are a necessary first step).

---

> > > ### Author Response · Authors · 2025-08-02
> > >
> > > We thank the reviewer for the kind and encouraging words, and for highlighting the importance of our contributions. We appreciate the emphasis on accessibility and will strive to improve the exposition accordingly. We are also grateful for the interest expressed in future extensions to deeper models.

---

### Official Review · Reviewer_JeQC · 2025-06-30

**Clarity:** 2
**Significance:** 4
**Originality:** 3
**Rating:** 5
**Confidence:** 3

**Summary:**

In this paper, the authors study the universality of equivariant neural nets, and particularly its relationship with their separation power, that is, their ability to distinguish elements in different equivalence class. After a (lengthy) introduction on the different notions, they exhibit an example of networks that have the same separation power for point clouds, but different universality (one is universal, the others are not). Their main theorems pertains to necessary/sufficient conditions for universality or failure thereof, and the vanishing of certain differential operators on classes of function.

**Questions:**

See above, mostly related to invariance vs. equivariance and other examples of "true" constrained separability (the given example uses maximal separability).

**Ethical Concerns:**

["NO or VERY MINOR ethics concerns only"]

**Final Justification:**

I keep my positive view of this paper, despite some heaviness in the notations.

**Limitations:**

Limitations are appropriate.

**Quality:**

3

**Strengths And Weaknesses:**

Strength
- this negative results are quite original in this active literature. I believe that, at least the concept, is an important step towards a better understanding of separation, universality, and the building of invariant/equivariant architectures with such properties.

Weaknesses
- the main weakness of the paper is that it is extremely notation-heavy and quite hard to parse. It' feels like each line introduces new notations up until the very end, making the results quite obfuscated and hard to judge. Examples, main theorems, etc. feel rushed: for instance, one of the main theorem is about differentiability, and this notion is never introduced before the theorem itself! I feel like this paper would really benefit from being journal length, or from being in a much more focused and example-heavy way.

- with so many notations, some of them clash, eg $M_d$ and $M_h$ with integers $d$ and $h$, and some are redundant, I believe neural nets are re-defined many times along the way. It's very easy to lose track.

- proposition 9 is the core of the idea and it is a bit confusing. width-1 convolutional networks are mixed with permutation-invariance (which is weird), and the proof is quite confusing, for such seemingly simple architectures. Examples like this should be more explicitely written.

- related to this, another weakness is that, as of now, all the notions and theorems feel a bit too "heavy" for the rather simple counter-example that is the core of the authors' demonstration. For instance, the authors introduce a quite convoluted notion of "separation class" and "separation-constrained universality", and then only apply it to maximal separation over point clouds. Could other, less trivial examples be found?

- The authors claim that equivariance can be summarized to invariance, but it seems a bit imprecise. They correctly that, *for negative results*, this is the case; of course failure to be invariant-universal is also failure to be equivariant-universal. But the converse is a bit brushed off.

- A large branch of proofs of universality precisely proves it by using separability + Stone-Weierstrass theorem. It could be useful to include it in a remark.

In conclusion, I believe this paper starts an important line of results, but would really benefits from being longer, with more examples that are less trivial, and more time to discuss the two "main" theorems, which are rushed at the end of the paper used notions that were not introduced in the rest of the paper.

---

> ### Author Rebuttal · Authors · 2025-07-29
>
> We thank the reviewer for the detailed comments and for recognizing the conceptual importance of our results. We agree with the suggestions regarding clarity and presentation, and we provide detailed responses below.
>
> > the main weakness of the paper is that it is extremely notation-heavy and quite hard to parse. It' feels like each line introduces new notations up until the very end, making the results quite obfuscated and hard to judge. Examples, main theorems, etc. feel rushed: for instance, one of the main theorem is about differentiability, and this notion is never introduced before the theorem itself! I feel like this paper would really benefit from being journal length, or from being in a much more focused and example-heavy way.
>
> We thank the reviewer for the comment and acknowledge the concern raised. The terseness was indeed because we were constrained by space. We are confident that a single additional page will be sufficient to present the essential missing material in a complete and accessible manner. Further technical developments, such as an extended treatment of ridge functions or detailed arguments relying on this theory, would not enhance the paper’s readability and, even in a journal version, would be more appropriately placed in an appendix.
> Moreover, we plan to include figures to visualize the constructions, clarify key examples, and add brief proof sketches to enhance comprehension, both in depth and breadth, as suggested by Reviewer Wpwm.
> We also believe that these results would be of interest to the NeurIPS community and that the timely dissemination enabled by the conference format would provide greater benefit to the community.
>
> > with so many notations, some of them clash, eg $M_d$ and $M_h$ with integers $d$ and $h$, and some are redundant, I believe neural nets are re-defined many times along the way. It's very easy to lose track.
>
> The reviewer is right. We will change the notation by clearly distinguishing between $M_d$, $M_h$, and $N_h$ (which is indeed defined differently from $M_h$).
>
> > proposition 9 is the core of the idea and it is a bit confusing. width-1 convolutional networks are mixed with permutation-invariance (which is weird), and the proof is quite confusing, for such seemingly simple architectures. Examples like this should be more explicitely written.
>
> We understand that the reviewer found these examples confusing. To address this, we now clarify the discussion by presenting it in two separate parts:
> - First, we agree that it may seem unusual to discuss convolutions in the context of permutation invariance rather than translation invariance, which is a weaker constraint. However, when the filter length is 1, the convolution reduces to a scalar multiple of the identity which is a permutation equivariant operation.
> By composing such layers with a final invariant layer, one obtains a permutation-invariant network, which is obviously also translation equivariant.
> Nonetheless, other translation-equivariant models, constructed using larger filters, do not necessarily preserve permutation equivariance.
> In Proposition 9 and 16, our focus is on constructing minimal working examples of separation-constrained universality failure; thus, restricting to permutation equivariance and filters of length 1 was a natural choice.
> Beyond the case of permutation equivariance, we emphasize that the characterization of universality classes provided in Theorem 13, which we regard as a central contribution of our work, applies to arbitrary equivariant layer spaces. This level of generality, however, comes at the expense of notational simplicity.
>
> - Second, we recognize that the entire proof of Proposition 9 may feel unexpectedly complex given the simplicity of the architectures involved.
> However, we emphasize that the most technically involved part of the proof lies in Proposition 41, whose complexity stems from the generality of the result, which applies to arbitrary layer spaces.
> Although Proposition 41 appears only in the appendix, we believe that its generality makes it an important contribution, even if technical. We will make this clearer and more explicit in the revision.
>
> > related to this, another weakness is that, as of now, all the notions and theorems feel a bit too "heavy" for the rather simple counter-example that is the core of the authors' demonstration. For instance, the authors introduce a quite convoluted notion of "separation class" and "separation-constrained universality", and then only apply it to maximal separation over point clouds. Could other, less trivial examples be found?
>
> The notation, while admittedly dense, is necessary to formally state and prove the general theory, in particular Theorem 13 and Theorem 18. We will ease the reader in by simplifying some of the notation in the revision.
> At present, we do not have less trivial examples beyond those already included. Nevertheless, we emphasize that the existing example is sufficient to give a negative answer to the broader question *“Does separation imply approximation?”*.
>
> > The authors claim that equivariance can be summarized to invariance, but it seems a bit imprecise. They correctly that, for negative results, this is the case; of course failure to be invariant-universal is also failure to be equivariant-universal. But the converse is a bit brushed off.
>
> We agree with the reviewer’s comment and will address it carefully in the revision, showcasing the following technical details:
>
> The converse implication of Remark 11 does not generally hold. It is possible to construct counterexamples in which the invariant setting fails to inform about to the equivariant case, although such examples are often somewhat contrived. To clarify this, we plan to include the following in the revision: let $V, W, Z$ be permutation representations of a group $ G $, and write $Z = Z^G \oplus Z'$, where $Z^G$ denotes the space of invariant vectors and $ Z' $ its direct complement. Consider the layer spaces $M = \mathrm{Aff}_G(V, W)$ and $N = \mathrm{Aff}_G(W, Z')$, and define the universality class $\mathcal{U} = \mathcal{U}(M, N)$. Then, by construction, the pullback map $\pi^* $ introduced in Remark 11 maps all functions in $\mathcal{U}$ to $\\{ 0 \\}$, regardless of the choice of $M$ and $W$. This means that different equivariant universality classes with outputs in $Z'$ are collapsed on $\\{ 0 \\}$ by projecting through $\pi^*$ onto $ Z^G $, hence indistinguishable from the invariant perspective.
>
> That said, we want to emphasize that our ridge-function-based formulation does extend to the equivariant setting and can be employed to tackle the example discussed above. We did not highlight this extension in the main text because it does not yield a clean analog of Theorem 13. Nevertheless, to clarify this point, we will expand on the following: Proposition 40 (L1017–L1030) generalizes to the equivariant setting with minor adaptations, namely by expressing the matrix $A$ in terms of a suitable basis of equivariant maps and adding a sum at L1025. As a result, the first part of the proof of Theorem 13 continues to apply, and a differential constraint still characterizes the class. However, the equivalence in lines L1066–L1067 no longer holds in the equivariant case, since $f$ is not assumed to be invariant. This leads to a more complex formulation that depends explicitly on the output representation and the structure of the final layer, which is why we chose to omit it in the current version.
>
> > A large branch of proofs of universality precisely proves it by using separability + Stone-Weierstrass theorem. It could be useful to include it in a remark.
>
> The reviewer is correct that a large portion of the literature focuses on separability combined with the Stone–Weierstrass theorem. In the revised version, we will add a dedicated remark discussing this perspective. Here, the key observation is that the classical Stone–Weierstrass argument does not apply here, as the function families under consideration, such as PointNets, do not form an algebra.

---

> > ### Comment · Reviewer_JeQC · 2025-08-04
> >
> > Thank you for your answers, which have largely alleviated my concerns.
> >
> > Regarding the Stone-Weierstrass remark, some specific variants have been proven for the equivariant case, see eg your reference [37] by Keriven and Peyré. I agree that it's probably not flexible enough to be useful here, but it still draws a classical link between separability and universality that could be mentioned in a discussion, along with why it does not apply here.
> >
> > I keep my positive score.

---

> > > ### Author Response · Authors · 2025-08-04
> > >
> > > We thank the reviewer for the positive feedback and for engaging in the discussion. We are glad that our responses have addressed the main concerns. We also appreciate the insightful remark regarding the Stone-Weierstrass theorem and reference [37] by Keriven and Peyré. We agree that this classical connection between separability and universality in the equivariant setting deserves mention, even if it does not directly apply to our context. We will include a brief discussion of this link in the revised version.

---

### Official Review · Reviewer_FKDG · 2025-07-01

**Clarity:** 2
**Significance:** 2
**Originality:** 2
**Rating:** 4
**Confidence:** 3

**Summary:**

The paper revisits the relationship between approximation power and separation power for equivariant neural networks. While most prior work focuses on separation power—the ability to distinguish inputs up to symmetry—the authors ask whether networks that separate equally well can also approximate equally well. By characterizing the universality classes of equivariant networks, it shows that separation power alone cannot fully describe approximation power: models with identical separation power may differ in their approximation ability. Particularly, it exhibits three popular set models whose separation power is maximal yet whose approximation power is strictly nested.

**Questions:**

See **Weaknesses**.

**Ethical Concerns:**

["NO or VERY MINOR ethics concerns only"]

**Final Justification:**

After reviewing the rebuttal and the other reviewers’ comments, I maintain my score and remain inclined toward acceptance.

**Limitations:**

Yes.

**Quality:**

3

**Strengths And Weaknesses:**

**Strengths**
- It was known that universal approximation implies maximal separation power. This work provides another piece, suggesting that the opposite is not true: maximal separation power does not necessarily lead to universal approximation.
- This theoretical finding implies that function approximation power and speration power of equivalence classes are fundamentally distinct. Therefore, model analysis and architectural design should address separation and approximation as distinct criteria, instead of relying on just one of them.



**Weaknesses**
- The main theoretical results are mainly restricted to shallow networks
- There are no empirical validations
- The notion is heavy to read

---

> ### Author Rebuttal · Authors · 2025-07-29
>
> We are grateful to the reviewer for engaging with our work and for recognizing the significance of distinguishing separation and approximation power. The points raised, particularly regarding scope and lack of experimental evaluation, are addressed in the following.
>
> > The main theoretical results are mainly restricted to shallow networks
>
> We certainly recognize this limitation and are actively working on extending our analysis to deeper architectures. However, we would like to take this opportunity to emphasize the motivation for our work. We were motivated by the following question *“Is separation a complete proxy for approximation?”* Our findings clearly show that it is not, which is the central message of the paper. We consider it to be a key jumping off point for the analysis of deep architectures.
>
> > There are no empirical validations
>
> We agree that experiments would help strengthen the soundness of the work. However, approximation cannot be directly controlled by the experimenter. It can only be indirectly influenced through factors such as hyperparameters or activation functions. These factors, however, affect not only approximation but also other critical aspects, such as generalization and trainability, which in turn impact empirical outcomes. As a result, designing experimental settings that isolate the effect of approximation power is highly challenging, since disentangling the influence of these interacting components is non-trivial.
> An example of a possible experiment would be attempting to approximate the multiplication function with a shallow PointNet. However, it would be difficult to determine whether any failure arises from the inherent approximation limitations identified in this paper or from limitations of the optimization procedure itself.
> Once we gain a clearer understanding of these interactions, we believe it will be possible to conduct a more detailed and meaningful experimental study. In alignment with the final part of the reviewer’s comment, we see our theoretically grounded contribution as an important first step toward understanding the learning pipeline of equivariant networks, providing a foundation for future results that can be empirically validated in a principled way.
> We will ensure to mention these points in the revised manuscript.
>
> > The notion is heavy to read
>
> We employ the notation introduced in Pacini et al., *Separation Power of Equivariant Neural Networks*. We understand that the notation can feel heavy, and we will make sure to address these issues by improving notation and presentation and adding illustrations as suggested by Reviewer Wpwm.

---

> > ### Comment · Reviewer_FKDG · 2025-08-05
> > **Official Comment by Reviewer FKDG**
> >
> > Thank you for the rebuttal. Most of my concerns are addressed, and I will maintain my score.

---

> > > ### Author Response · Authors · 2025-08-06
> > >
> > > We thank the reviewer for the positive assessment and constructive comments.

---

### Official Review · Reviewer_Ywy9 · 2025-07-02

**Clarity:** 3
**Significance:** 2
**Originality:** 3
**Rating:** 5
**Confidence:** 2

**Summary:**

Observes that we should care about more than universality “up-to-separation” of equivariant neural networks, we should also care about their capacity to approximate a function. Characterize universality for shallow models. Show that there are families of models (e.g. shallow PointNets and CNNs with a filter width of 1) that differ in their approximation capacity but have identical ability to separate inputs mod symmetry.

**Questions:**

- For Thm 18, Stone–Weierstrass also requires point-separation?
- The n=3 PointNet case seemed rush. Could you perhaps add a bit more detail here.

**Ethical Concerns:**

["NO or VERY MINOR ethics concerns only"]

**Limitations:**

yes

**Quality:**

4

**Strengths And Weaknesses:**

## Strengths

Offers an interesting, novel perspective on the universality of equivariant networks. Construct real examples where separation and universality differ. The proofs are relatively clear and the assumptions are generally stated.

## Weaknesses

- Importance: the theorems target depth-2 models. It seems non-trivial to extend to deep architectures (and motivate this distinction), in-part because compositions of equivariant layers have significant approximation capacity.
- Similarly: is there a reason why universality captures certain intuitions better than separability and why should this matter?
- Minor: In the conclusion, you state that you formulate shallow invariant networks as superpositions of ridge functions. You do not explicitly connect these in the main body/cite your appendix.

---

> ### Author Rebuttal · Authors · 2025-07-29
>
> We thank the reviewer for the thoughtful review and for highlighting the novelty of our perspective on universality beyond separation. The comments provided have helped us refine the presentation and clarify key aspects of our framework. Our responses follow below.
>
> > Importance: the theorems target depth-2 models. It seems non-trivial to extend to deep architectures (and motivate this distinction), in-part because compositions of equivariant layers have significant approximation capacity.
>
> We acknowledge this limitation and are in fact actively working on extending our analysis to deeper architectures. While additional examples and a more comprehensive theoretical treatment for deep networks would indeed strengthen the work, we emphasize that the examples presented already suffice to tackle the key conceptual question: “Can separation alone fully capture approximation?” Our results clearly demonstrate that the answer is no, which constitutes the main message of the paper. We consider it to be a key conceptual milestone before tackling the analysis for deeper architectures.
>
> > Similarly: is there a reason why universality captures certain intuitions better than separability and why should this matter?
>
> We thank the reviewer for highlighting this important point. As demonstrated in our paper, separability is only a necessary condition for universality but not a sufficient one. But, for learning, universality is necessary to some extent. Indeed, as we show in the paper (L339–356), shallow PointNets cannot approximate, and therefore cannot learn, even a relatively simple function such as multiplication. This strongly suggests that more complex and realistic functions are also beyond the approximation (and learning) capacity of such models. Our goal in this paper is precisely to emphasize that this limitation cannot be resolved by considering separability alone.
>
> > Minor: In the conclusion, you state that you formulate shallow invariant networks as superpositions of ridge functions. You do not explicitly connect these in the main body/cite your appendix.
>
> We thank the reviewer for bringing this to our attention. We agree that explicitly referencing the appendix where appropriate will improve cohesion across sections. This was an oversight, and we will ensure these references are clearly added.
>
> > For Thm 18, Stone–Weierstrass also requires point-separation?
>
> The reviewer is correct that there are multiple formulations of the Stone–Weierstrass theorem, many of which require point separation. In our case, we employ a formulation that is separation-aware, as found in DeVore’s *Constructive Approximation* (Theorem 4.3). Moreover, note that condition (a) is always satisfied here, since $g(x) = 0$ does not hold for all neural networks due to the presence of bias terms in the output layer. We will make this reasoning more explicit in the revision.
>
> > The n=3 PointNet case seemed rush. Could you perhaps add a bit more detail here.
>
> We thank the reviewer for pointing out that this case felt rushed. We will expand this computation in the revised version to make the argument clearer and more complete.

---

> > ### Comment · Reviewer_Ywy9 · 2025-08-03
> >
> > Thanks for your detailed reply. My concerns are addressed and I will maintain my score.

---

> > > ### Author Response · Authors · 2025-08-04
> > >
> > > We thank the reviewer for the positive feedback and are glad that our reply has addressed the concerns. We appreciate the time and attention dedicated to the review process.

---

### Official Review · Reviewer_Wpwm · 2025-07-04

**Clarity:** 3
**Significance:** 3
**Originality:** 3
**Rating:** 5
**Confidence:** 4

**Summary:**

This paper introduces a theoretical framework to characterize the universality of shallow equivariant neural networks, without assuming universal components like MLPs. For this, concepts of separation power of equivariant networks are leveraged and links between separation power and universality and obtained. Necessary criteria for universality are derived, along with examples of applications of these ideas to widely used architectures.

**Questions:**

- One component that I felt would make the story more complete for me is a characterization of conditions for $C_\rho = C_G$, e;.g. when does separation constrained universality coincide with universality? I think this would help in understanding the implications of Theorem 18. Are results from [6] enough for this? If yes, it would be nice to include them in the discussion.
- If I understand correctly, the framework does not fully generalize to the equivariant case, since only non-universality (and not universality) transfers from invariance to equivariance. Is that right? If yes, this should be mentioned more explicitely.

**Ethical Concerns:**

["NO or VERY MINOR ethics concerns only"]

**Final Justification:**

I acknowledge the authors rebuttal and am looking forward to see the paper with the promised changes. I am happy to recommend this paper for acceptance.

**Limitations:**

Limitations are discussed.

**Paper Formatting Concerns:**

None.

**Quality:**

4

**Strengths And Weaknesses:**

## Strengths
- The paper is well-written and overall a great read despite the heaviness of the theoretical framework and notations
- The results of the paper generalize and extend several existing results in a non-trivial and very interesting way. They have practical implications and can be used to show explicitly if some architectures satisfy universal criteria. As such they will be useful and important to the geometric deep learning community
- The limitations of the analysis are clearly stated

## Weaknesses
- Some related works are not mentioned, notably [1,2], which formalize concepts related to separation constrained universality for graphs. Another line of work sidesteps the issues of expressivity of equivariant neural networks by adaptation of universal architectures [3,4] and is worth mentioning.
- The paper is clearly written, but understanding could potentially be improved by including some figures (they could fit with the additional page or in appendix). One recommendation is for example to represent the constructions for the layer spaces (equation 1) using coloured bipartite graphs as is done for example in [5] (This paper is also worth citing as it introduced a formulation close to equation 1). I think this would allow to visualize all the examples provided in the paper in an intuitive way.
- The paper would also be improved if the authors could add short proof sketches and find some intuition that is captured by each of the main theorems. In their current form the statements are a bit dry.

[1] Azizian and Lelarge, Expressive Power of Invariant and Equivariant Graph Neural Networks, 2021

[2] Chen et al., On the equivalence between graph isomorphism testing and function approximation with gnns, 2019

[2] Puny et al., Frame averaging for invariant and equivariant network design, 2021

[3] Kaba et al., Equivariance with learned canonicalization functions, 2022

[5] Ravanbakhsh et al. Equivariance through parameter-sharing, 2017

[6] Pacini et al. Separation Power of Equivariant Neural Networks, 2025

---

> ### Author Rebuttal · Authors · 2025-07-29
>
> We thank the reviewer for the positive and encouraging feedback. We are pleased that the paper was found to be well-written and that its conceptual and technical contributions were appreciated. We address the reviewer’s constructive suggestions and questions below.
>
> > Some related works are not mentioned, notably [1,2], which formalize concepts related to separation constrained universality for graphs. Another line of work sidesteps the issues of expressivity of equivariant neural networks by adaptation of universal architectures [3,4] and is worth mentioning.
>
> We thank the reviewer for highlighting the missing references. We agree that these papers are very relevant to the discussion on equivariant universality, and their inclusion will strengthen our paper. We will incorporate them in the revised version and add commentary on how they relate to our framework.
>
> > The paper is clearly written, but understanding could potentially be improved by including some figures (they could fit with the additional page or in appendix). One recommendation is for example to represent the constructions for the layer spaces (equation 1) using coloured bipartite graphs as is done for example in [5] (This paper is also worth citing as it introduced a formulation close to equation 1). I think this would allow to visualize all the examples provided in the paper in an intuitive way.
>
> We appreciate the suggestion and will incorporate it in the revision. We agree that adding figures will enhance the paper’s readability. In particular, we will include several visual representations of the layer space constructions appearing in Proposition 9, using colored bipartite graphs, as suggested. With the additional page allowance, if the paper is accepted, we can also illustrate key examples more intuitively, helping to clarify the more notation-heavy sections.
>
> > The paper would also be improved if the authors could add short proof sketches and find some intuition that is captured by each of the main theorems. In their current form the statements are a bit dry.
>
> We appreciate the suggestion to add brief proof sketches and intuitive explanations for the main theorems. We do recognize that the current presentation may come across as overly dry. We were constrained by space and hence the concision. In the revised version, using the additional page, if the paper is accepted, we will summarize the key ideas underlying each result, with particular emphasis on the role of the differential operators and their connection to the theory of ridge functions. This should make the results more accessible without requiring extensive background on ridge functions, which could risk making the main points more cumbersome.
>
> > One component that I felt would make the story more complete for me is a characterization of conditions for $C_\rho = C_G$, e.g., when does separation constrained universality coincide with universality? I think this would help in understanding the implications of Theorem 18. Are results from [6] enough for this? If yes, it would be nice to include them in the discussion.
>
> The reviewer is correct that, for the invariant case, $C_\rho = C_G$ is equivalent to $\rho$ being equal to the orbits. Determining when a neural space with separation power $\rho$ yields $C_\rho = C_G$ can be understood via Theorem 1 in [6]. Although characterization may require difficult computations depending on subtle structural constraints, the difficulty arises from the complexity of the setting and the presence of multiple factors that can influence separation power. To clarify this point, we will expand on the example of $S_n/A_n$ as the input representation and $S_n/H$ with $H < A_n$ for the hidden representation. This has been only partially discussed after Theorem 18 (L364-368).
> In this case, we can prove that $\mathcal U (\mathbb R^{S_n/A_n}, \mathbb R^{S_n/A_n}, \mathbb R) = \mathcal U (\mathbb R^{S_n/A_n}, \mathbb R^{S_n/H}, \mathbb R) = \mathcal C_{S_n}(\mathbb R^{S_n/A_n})$.
> In particular, $\rho (\mathcal U (\mathbb R^{S_n/A_n}, \mathbb R^{S_n/A_n}, \mathbb R)) = \rho (\mathcal U (\mathbb R^{S_n/A_n}, \mathbb R^{S_n/H}, \mathbb R))$.
> This family of examples illustrate how different representations can induce the same separation power, and how this also depends on the input representation.
> It points to a highly non-homogeneous, asymmetric, and somewhat counterintuitive landscape in which architectural choices influence separability.
> Obtaining a characterization of when $C_\rho = C_G$ that is significantly simpler than the one provided in Theorem 1 of [6] may indeed be infeasible. We will ensure that these observations are appropriately integrated in the revised manuscript.
>
> > If I understand correctly, the framework does not fully generalize to the equivariant case, since only non-universality (and not universality) transfers from invariance to equivariance. Is that right? If yes, this should be mentioned more explicitely.
>
> Thanks for the incisive comment. If referring to Remark 11, the reviewer’s understanding is correct. Examples showing that universality does not extend from the invariant case to the equivariant case can indeed be constructed, although in our opinion they tend to be somewhat artificial. We will clarify this by adding the following counterexample and subsequent discussion:
>
> Let $V$, $W$, $Z$ be permutation representations of $G$, let $Z^G$ be the trivial subspace of $Z$ and $Z’$ be its complement, i.e., $Z = Z^G \oplus Z’$. Define $M = \text{Aff}_G(V, W)$ and $N = \text{Aff}_G(W, Z’)$ and consider the universality class $\mathcal U = \mathcal U(M, N)$. In this case, the pullback map $\pi^*$ defined in Remark 11 always projects $\mathcal U$ to $\\{0\\}$ independently of the choice of $M$ and $W$. Therefore two equivariant universality classes $\mathcal U \subsetneq \mathcal V$ with output layer $N$ would be projected to the same invariant class $\\{0\\}$.
>
> However, we will also underline how the ridge-function framework does transfer to the equivariant case. We have not emphasized this result because it is difficult to formulate a closed-form characterization analogous to Theorem 13. However, to better address this question, we find it relevant to highlight the following remarks, which we will also clarify in the revision.
>
> Proposition 40 (L1017-1030) transfers without much overhead to the equivariant case by decomposing $A$ through some basis of linear maps and adding another sum in the formula at L1025.
> Therefore, the first part of the proof of Theorem 13 also transfers to the equivariant case and the differential constraint is maintained, but biimplications at L1066-1067 do not transfer since we are not assuming $f$ to be invariant. Hence, leading to a more cumbersome formulation that we omitted due to its strong dependence on representations and layer spaces.

---

> > ### Comment · Reviewer_Wpwm · 2025-08-01
> >
> > I acknowledge the authors rebuttal and am looking forward to see the paper with the promised changes. I am happy to recommend this paper for acceptance.

---

> > > ### Author Response · Authors · 2025-08-02
> > >
> > > We thank reviewer for the constructive engagement and insightful suggestions, which will greatly improve the clarity and presentation of the paper.

---

### Decision · Program_Chairs · 2025-09-17

**Decision:**

Accept (spotlight)

**Comment:**

The paper theoretically studies the expressive power of equivariant neural networks, presenting several important results. The reviewers appreciated this contribution and assigned positive scores. I am happy to recommend acceptance as a spotlight presentation.